# Upregulated pexophagy limits the capacity of selective autophagy

Kyla Germain[1,2], Raphaella W. L. So [2,3], Laura F. DiGiovanni[1,2], Joel C. Watts [2,3], Robert H. J. Bandsma [4,5] ✉ & Peter K. Kim [1,2] ✉

Selective autophagy is an essential process to maintain cellular homeostasis through the constant recycling of damaged or superfluous components. Over a dozen selective autophagy pathways mediate the degradation of diverse cellular substrates, but whether these pathways can influence one another remains unknown. We address this question using pexophagy, the autophagic degradation of peroxisomes, as a model. We show in cells that upregulated pexophagy impairs the selective autophagy of both mitochondria and protein aggregates by exhausting the autophagy initiation factor, ULK1. We confirm this finding in cell models of the pexophagy-mediated form of Zellweger Spectrum Disorder, a disease characterized by peroxisome dysfunction. Further, we extend the generalizability of limited selective autophagy by determining that increased protein aggregate degradation reciprocally reduces pexophagy using cell models of Parkinson's Disease and Huntington's Disease. Our findings suggest that the degradative capacity of selective autophagy can become limited by an increase in one substrate.

Macroautophagy (hereafter referred to as autophagy) is a conserved process responsible for the breakdown of cytoplasmic material, including dysfunctional or superfluous organelles[1]. Autophagy is mediated by a double-membraned vesicle called an autophagosome that first sequesters cytoplasmic material and later fuses with a lysosome for degradation[1]. Selective autophagy occurs when a particular cytoplasmic substrate is designated for degradation by an accumulation of specialized autophagy receptors on its outer surface[1]. In mammalian cells, autophagy receptors either target directly to substrates upon expression or bind to an "eat me" signal that is tagged on substrates, most commonly ubiquitin[2].

At least 13 distinct selective autophagy pathways have been described in mammals, including the degradation of peroxisomes (pexophagy), mitochondria (mitophagy), and protein aggregates (aggrephagy)[1]. Although differing in their mechanisms of activation, they all share the same machinery that forms the sequestering membrane. However, as selective autophagy is largely studied as individual events by inducing targeted organellar damage, it is unknown whether different selective autophagy pathways influence one another. And far less is known about the regulation of selective autophagy in conditions when multiple substrate pathways are simultaneously activated.

Zellweger Spectrum Disorder (ZSD) is a group of rare and inherited diseases characterized by a loss of peroxisomes leading to a multisystemic disorder that includes neurodegeneration[3]. ZSD is caused by mutations in one of 13 Peroxin (PEX) genes that are required for peroxisome maintenance[3]. Cellularly, the loss of peroxisomes in humans leads to increased damaged mitochondria, abnormal lysosomes, and lipid droplets[4–7]. Studies of three mouse models of ZSD revealed an accumulation of autophagy substrates, including dysfunctional mitochondria and alpha-Synuclein (αS) oligomers in the liver and brain[8–10]. However, it is not understood why ZSD cells cannot clear these cytotoxins, despite no apparent defects in autophagy.

Previously, the PEX protein, PEX13, was proposed to regulate mitophagy and virophagy as depletion of this gene impaired both

[1]Cell Biology Program, The Hospital for Sick Children, Toronto, ON M5G 1X8, Canada. [2]Department of Biochemistry, University of Toronto, Toronto, ON M5S 1A8, Canada. [3]Tanz Centre for Research in Neurodegenerative Diseases, Toronto, ON M5T 0S8, Canada. [4]Translational Medicine Program, The Hospital for Sick Children, Toronto, ON M5G 1X8, Canada. [5]Department of Nutritional Sciences, Faculty of Medicine, University of Toronto, Toronto, ON M5S1A8, Canada. ✉e-mail: robert.bandsma@sickkids.ca; pkim@sickkids.ca

selective autophagy pathways[11]. However, we have recently demonstrated that the loss of PEX13 causes an upregulation of pexophagy in cells, suggesting the possibility that pexophagy and not the loss of PEX13 may impair other selective autophagy pathways[12]. Similar upregulation in pexophagy occurs in cells depleted of PEX1, the most commonly mutated gene in ZSD[13]. Given that distinct selective autophagy pathways require overlapping machinery, this raises the possibility that increased pexophagy may limit the activity of other selective autophagy pathways in ZSD.

In this study, we examine whether distinct selective autophagy pathways can influence one another. We report that cells with upregulated pexophagy have impaired aggrephagy and mitophagy. Impaired selective autophagy was not a result of the loss of peroxisomes or their function, but instead, it was due to saturation of the autophagic capacity of the cell at basal state by the upregulated pexophagy activity. Further, our studies identify Unc-51-like kinase 1 (ULK1) as a limited component that restricts selective autophagy in cells with upregulated pexophagy. Additionally, we provide data to support the generalizability of our findings using cell models of Parkinson's and Huntington's Disease. Together, our studies show that pexophagy-dependent limitation of mitophagy and aggrephagy contributes to pathological protein aggregation and mitochondrial dysfunction in models of ZSD, demonstrating that one form of selective autophagy can influence others.

## Results

### Depletion of PEX1 or PEX13 induces pexophagy

To investigate whether one substrate of selective autophagy can influence the degradation of another, we used a cell model where pexophagy was perpetually upregulated. We recently demonstrated that the depletion of PEX1 or PEX13, two components of peroxisomal matrix import machinery, causes a buildup of ubiquitinated PEX5 on the peroxisomal membrane that triggers ubiquitin-dependent pexophagy (Fig. 1a)[12,13]. Since these two genes are not required for the formation of peroxisome membranes or membrane protein import, their loss results in the continuous degradation of old and new peroxisomes without causing significant cell death[12,13]. Expectedly, the depletion of PEX1 or PEX13 in HeLa (Fig. 1b–d) and HEK293 cells (Supplementary Data Fig. 1a, b) resulted in 20–40% peroxisome loss, as determined by quantification of the punctate structures visualized by immunofluorescent staining of the peroxisomal membrane protein PMP70. Whereas, depletion of another peroxisome matrix import factor, PEX14, did not affect the number of PMP70 punctate structures in HeLa or HEK293 cells (Fig. 1, Supplementary Data Fig. 1a, b). The loss of peroxisomes observed in PEX1 or PEX13 depleted cells was autophagy dependent, as co-depletion of the autophagy factor ATG12 prevented peroxisome loss as previously described (Fig. 1c, d, Supplementary Data Fig. 1a, b)[12,13].

### Increased pexophagy impairs aggrephagy

Using our model for upregulated pexophagy, we first asked whether increased pexophagy influences the degradation of protein aggregates via aggrephagy. Specifically, we examined the autophagic clearance of aggresome-like inducible structures (ALIS)[14] from HeLa cells following proteotoxic stress. Aggrephagy was assessed by monitoring the volume of ALIS in cells visualized with an FK2 antibody that binds to poly- and monoubiquitinated proteins, but not free ubiquitin. Proteotoxic stress with 3-h puromycin treatment resulted in a robust production of ALIS (Fig. 1e–g). In control cells transfected with either no siRNA (mock) or non-targeting (control) siRNA, ALIS volume was largely reduced following a 5-h washout/clearance period, as indicated by a significant decline in FK2-aggregate volume (Fig. 1e–g). Whereas, depletion of ATG12 impaired ALIS clearance in line with previous reporting that puromycin-induced ALIS are degraded by autophagy (Fig. 1f, g)[14–17].

To investigate whether increased pexophagy influences aggrephagy, we examined ALIS clearance in cells depleted of either PEX1 or PEX13. Depletion of PEX1 and PEX13 showed no differences in the formation of ALIS compared to cells treated with control or PEX14-targeting siRNA (Fig. 1f, g). However, unlike these controls, cells with increased pexophagy were unable to eliminate aggregates during the 5-h clearance period (Fig. 1f, g). We further quantified aggrephagy activity by calculating the percentage clearance of ALIS compared to mock treated cells and observed that cells depleted of PEX1 or PEX13 showed a similar decrease in aggrephagy activity as ATG12 depleted cells, suggesting impaired aggrephagy (Fig. 1g).

### PEX13 or PEX1 are not required for aggrephagy

It was previously postulated that PEX13 was required for selective autophagy as the loss of PEX13, but not other peroxisome biogenesis factors PEX14 or PEX19, prevented mitophagy and virophagy in cultured cells[11]. To test whether PEX13 is required for aggrephagy, instead of influencing aggrephagy via increased pexophagy, we co-depleted HeLa cells of the peroxisomal E3 ubiquitin ligase, PEX2, that is required for the induction of pexophagy (Fig. 1a, h)[12,18]. We corroborated previous reporting that co-depletion of PEX13 and PEX2 rescues the peroxisome loss observed in cells depleted of PEX13 alone, indicating that PEX2 co-depletion is sufficient to inactivate pexophagy in PEX13-depleted cells (Fig. 1i). Co-depletion of PEX2 with PEX13 also rescued aggrephagy activity compared to cells depleted of PEX13 alone, supporting that the upregulation of pexophagy and not the loss of PEX13 function impaired aggrephagy (Fig. 1j).

To further validate that upregulated pexophagy and not the loss of PEX13 or PEX1 is responsible for decreasing aggrephagy activity, we induced pexophagy without depleting any peroxisomal genes. It was previously shown that targeting a ubiquitin motif to the cytosolic face of the peroxisomal membrane by ectopically expressing PMP34-GFP-UBKo can induce pexophagy[19]. PMP34-GFP-UBKo is a chimera consisting of the peroxisomal membrane protein, PMP34, fused to a GFP and a ubiquitin moiety whose lysine residues were mutated to arginine to prevent polyubiquitination[19]. As previously shown[19], expression of PMP34-GFP-UBKo resulted in significant peroxisome loss compared to cells expressing the non-ubiquitin tagged PMP34-GFP (Fig. 2a, b). When subjected to the puromycin-ALIS aggrephagy assay, cells expressing PMP34-GFP-UBKo had less aggrephagy activity compared to PMP34-GFP expressing cells (Fig. 2c–e), similar to the results observed in PEX1 and PEX13-depleted cells (Fig. 1g).

We observed partial colocalization of PMP34-GFP-UBKo and ALIS following proteotoxic stress, suggesting that peroxisomes and ALIS may be clustered to allow for autophagy. Since ALIS were visualized with the anti-polyubiquitin antibody, FK1, labeling of the mono-ubiquitin motif on PMP34-GFP-UBKo should be subverted. However, to ensure that our quantification of FK1 represented only ALIS and not ubiquitinated peroxisomes, we subtracted the volume of PMP34-GFP-UBKo structures from the volume of FK1-labeled structures per cell (Fig. 2d). When adjusted for the PMP34-GFP-UBKo structures, we still observed less aggrephagy activity in cells expressing PMP34-GFP-UBKo compared to those expressing PMP34-GFP (Fig. 2e).

Finally, we tested whether the loss of functional peroxisomes may be responsible for impairing aggrephagy by measuring ALIS clearance in a PEX19 deficient ZSD human fibroblast line, PBD399-T1[20]. PEX19 is an essential peroxisome biogenesis factor, wherein its absence peroxisome biogenesis is abolished[20]. While PBD399-T1 cells lack peroxisomes as indicated by the lack of PMP70, they still express some peroxin proteins (Supplementary Data Fig. 2a, b). Therefore, we treated the cells with siRNA to ensure a loss of PEX1 and PEX13 (Supplementary Data Fig. 2a). Despite lacking peroxisomes, PBD399-T1 cells were able to clear puromycin-induced ALIS, supporting their capacity for aggrephagy, while disrupting autophagy by ATG12 depletion prevented ALIS clearance (Supplementary Data Fig. 2c–e). Collectively,

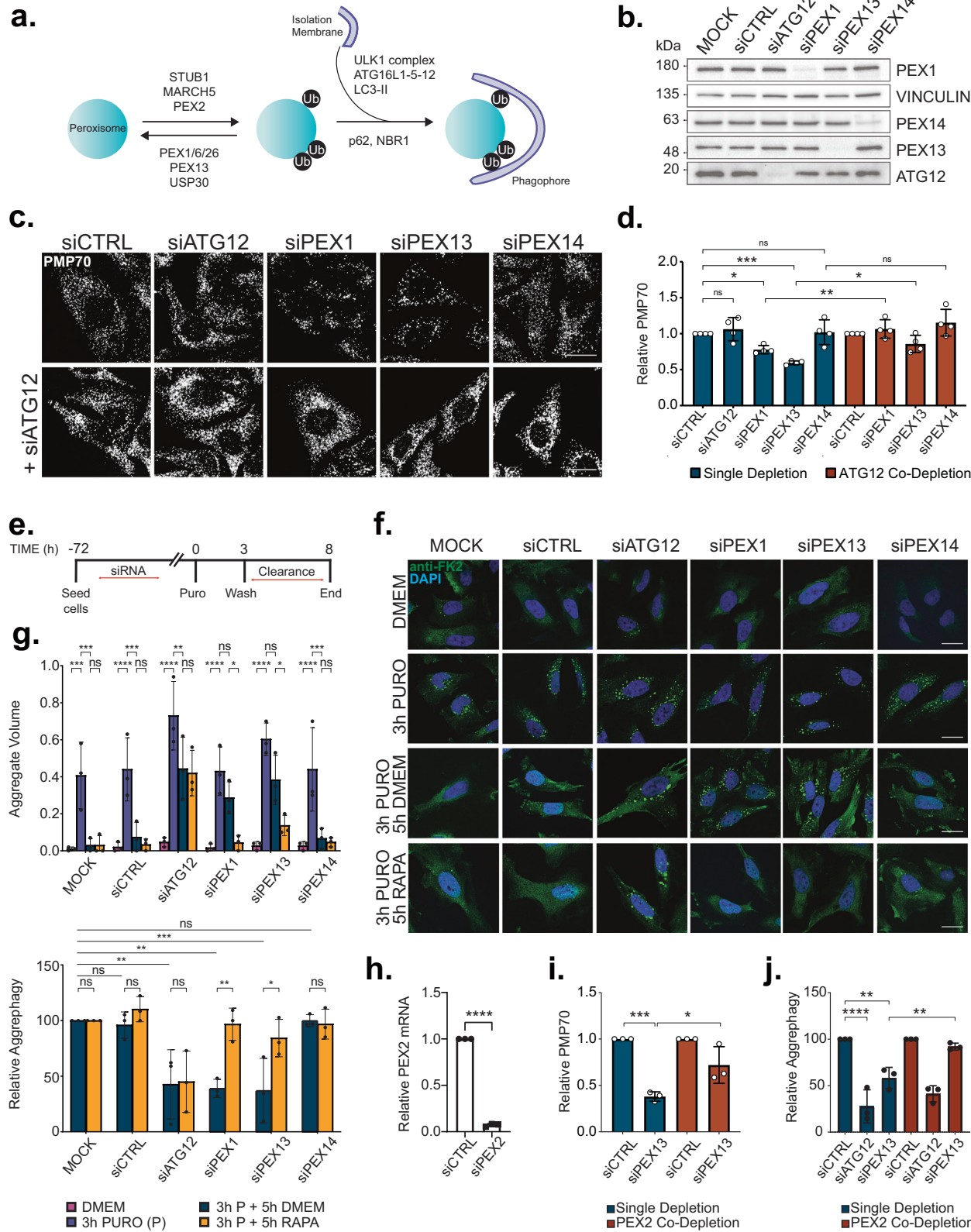

these data indicate that upregulation of pexophagy and not the loss of PEX1, PEX13, or peroxisome function elicit impaired aggrephagy.

## Upregulating autophagosome formation rescues aggrephagy

Mechanistic target of rapamycin complex 1 (mTORC1) is a major regulator of autophagy, limiting the amount of autophagosomes that a cell can form by inhibiting the autophagosome biogenesis initiator,

ULK1[21–23]. Impaired aggrephagy in conditions of increased pexophagy suggests that peroxisomes are consuming the available autophagosome biogenesis machinery, thus limiting aggrephagy from occurring. To test this hypothesis, we treated cells with the mTORC1 inhibitor, rapamycin, during the 5-h clearance period of our aggrephagy assays to upregulate autophagosome formation and autophagy flux. We confirmed that 5-h of rapamycin treatment inhibited mTORC1 activity

**Fig. 1 | PEX1 or PEX13 depletion impairs puromycin-induced aggrephagy.**
**a** Schematic of mammalian ubiquitin-dependent pexophagy. **b** Immunoblot of HeLa cells treated with siRNA and probed for the indicated proteins.
**c** Representative images of HeLa cells treated with the indicated siRNA and immunostained for the peroxisomal marker, PMP70. Scale bar, 25 μm.
**d** Quantification of PMP70 in (**c**) relative to siCTRL conditions. PMP70 was measured by dividing the number of PMP70 puncta by cell volume (see Methods).
**e** Schematic of aggrephagy assay. HeLa cells were treated with the indicated siRNA prior to aggrephagy induction (see Methods). **f** Representative images from cells at each stage in the assay: DMEM, 3-h 5 μg mL$^{-1}$ Puromycin, 3-h 5 μg mL$^{-1}$ Puromycin followed by 5-h clearance period in DMEM, or 3-h 5 μg mL$^-$ Puromycin followed by 5-h clearance period in 2 μM Rapamycin. Cells were immunostained with the ubiquitin antibody FK2; blue=DAPI. Scale bars, 25 μm. **g** Quantification of aggrephagy

assay in (**f**). Aggregate Volume was calculated by dividing the total volume of FK2 puncta by cell volume. Aggrephagy activity was measured as the % clearance of FK2 aggregates, relative to mock transfected "3 h P + 5 h DMEM" cells. **h** PEX2 mRNA levels measured by qPCR in cells treated with control non-targeting or PEX2-targeting siRNA, relative to control. **i** Quantification of PMP70 in HeLa cells treated with the indicated siRNAs. **j** Aggrephagy activity in HeLa cells treated with the indicated siRNAs, relative to control siRNA-treated condition. Data are displayed as means ± standard deviation from *n* = 4 (d) or *n* = 3 (g, h, i) independent experiments. Statistical significance is denoted as *$P \leq 0.05$, **$P \leq 0.01$, ***$P \leq 0.001$, ****$P \leq 0.0001$, ns not significant. **d, i** One-way ANOVA, Tukey's multiple comparisons test. **g, j** Two-way ANOVA, Tukey's multiple comparisons test. **h** Unpaired *t*-test, two tailed. Source data and exact *P* values are provided as a Source Data file.

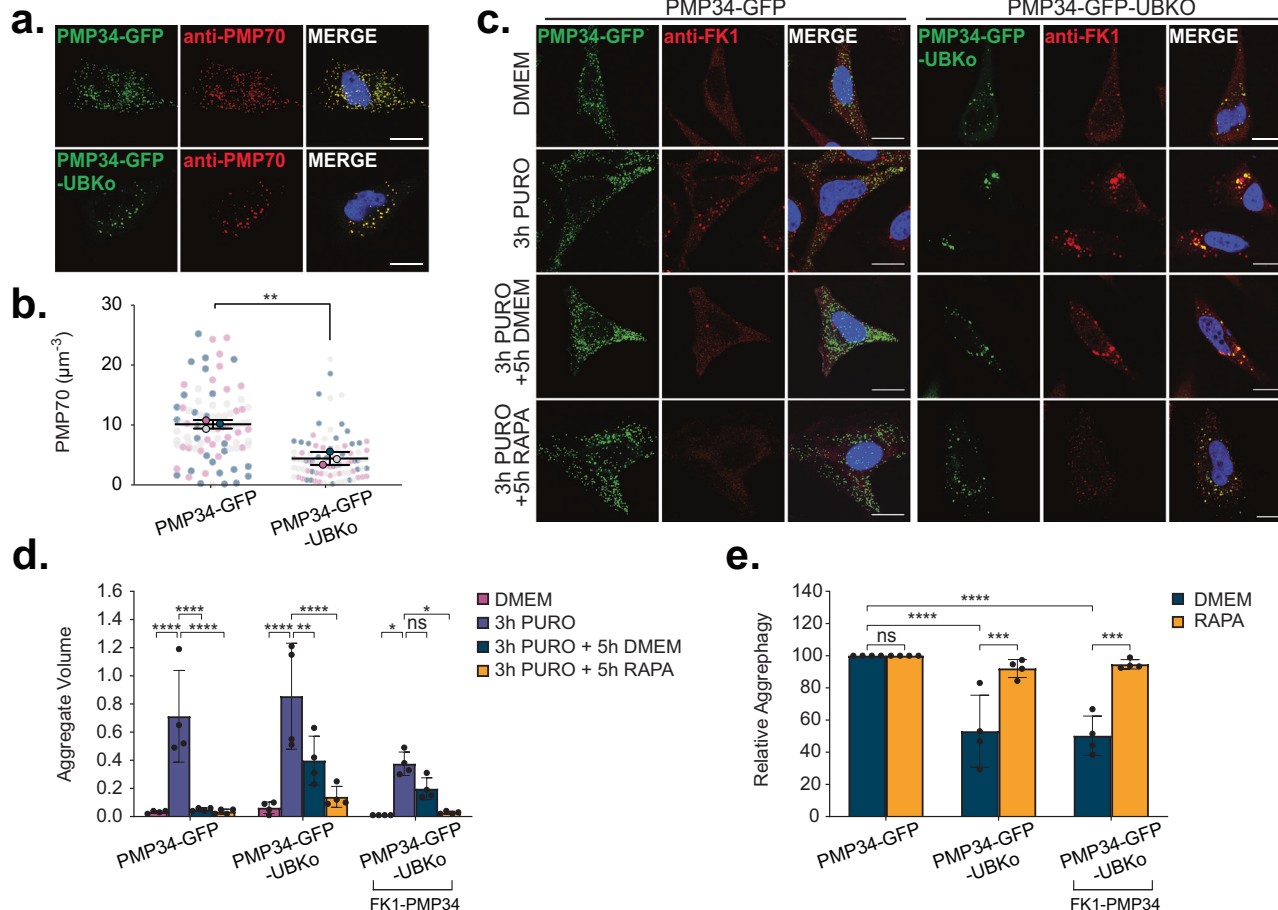

**Fig. 2 | PMP34-GFP-UBKo expression impairs puromycin-induced aggrephagy.**
**a** HeLa cells transfected with either PMP34-GFP or PMP34-GFP-UBKo 24-h before fixation and immunostaining for PMP70; blue = DAPI. Scale bars, 20 μm.
**b** Quantifications of PMP70 in (**a**). PMP70 was calculated by dividing the number of PMP70 puncta by cell volume (see Methods). **c** HeLa cells transfected with either PMP34-GFP or PMP34-GFP-UBKo 24-h before aggrephagy induction. Representative images from cells at each stage in the assay: DMEM, 3-h 5 μg mL$^{-1}$ Puromycin, 3-h 5 μg mL$^{-1}$ Puromycin followed by 5-h clearance period in DMEM, or 3-h 5 μg mL$^{-1}$ Puromycin followed by 5-h clearance period in 2 μM Rapamycin DMEM. Cells were immunostained for FK1; blue=DAPI. Scale bars, 25 μm. **d** Quantification of

Aggregate Volume in (**c**). Aggregate Volume was calculated by dividing the total volume of FK1 puncta by cell volume. **e** Quantification of aggrephagy activity in (**c**), relative to PMP34-GFP. Aggrephagy activity was measured as the % clearance of FK1 aggregates. Data are displayed as means (**d, e**) or individual cells overlaid with means, where different colors correspond to independent trials (**b**) ± standard deviation from *n* = 3 (**b**) or *n* = 4 (**d, e**) independent trials. Statistical significance is denoted as *$P \leq 0.05$, **$P \leq 0.01$, ***$P \leq 0.001$, ****$P \leq 0.0001$, ns not significant.
**b** Unpaired *t*-test, two tailed, performed with means. **d, e** Two-way ANOVA, Tukey's multiple comparisons test. Source data and exact *P* values are provided as a Source Data file.

in PEX1 or PEX13 cells by measuring phosphorylation of its downstream target, p70 S6 Kinase (p70-S6K) (Supplementary Data Fig. 3). When treated with rapamycin during the clearance period of the ALIS assay, we observed a significant improvement in aggrephagy activity in both PEX1 and PEX13, but not ATG12 depleted cells (Fig. 1e−g). Akin to the PEX1 and PEX13-depleted cells, rapamycin treatment improved the clearance of ALIS from PMP34-GFP-UBKo-expressing cells (Fig. 2c−e).

We next asked whether the loss of PEX1 or PEX13 may be reducing autophagy activity by activating mTORC1. However, we observed no significant difference in the phosphorylation state of mTORC1 and its downstream substrate p70-S6K between the various siRNA treatments suggesting that mTORC1 activity is not affected in cells depleted of PEX1 or PEX13 (Supplementary Data Fig. 3). Taken together with the rapamycin studies, these results indicate

that increased pexophagy limits aggrephagy without modulating mTORC1.

## Increased pexophagy impairs Parkin-dependent mitophagy

We next addressed whether increased pexophagy influences mitophagy. Mitophagy can be induced through multiple pathways. However, they can be grouped into either ubiquitin-independent or ubiquitin-dependent pathways[1], with the PINK1-Parkin mediated ubiquitin-dependent pathway being the most studied. Therefore, we first examined the effects of increased pexophagy on Parkin-mediated mitophagy induced with the mitochondrial respiration inhibitors oligomycin and antimycin A1 (OA)[24,25], in HEK293 cells stably expressing GFP-Parkin. These cells were used as OA treatment results in robust and accelerated mitophagy dynamics[26]. In mock or control siRNA-treated cells, mitochondrial clearance was observed following 8-h of OA treatment, as assessed by immunoblot of outer mitochondrial membrane (OMM; MFN2), inner mitochondrial membrane (IMM; CVα) and matrix proteins (Hsp60) (Fig. 3a–d). Inhibition of autolysosome fusion and degradation with bafilomycin A1 (BafA1) or depletion of the autophagy factor ATG12 did not prevent the degradation of MFN2, supporting its proteasomal degradation[1]. However, they did prevent the degradation of IMM and matrix proteins, confirming their autophagic degradation in control conditions (Fig. 3a–d). Mitochondria loss by autophagy following 8-h OA treatment was also confirmed with immunofluorescent imaging and intensity quantifications of CVα (Fig. 3e, f).

To examine whether pexophagy can influence mitophagy, we depleted cells of PEX1 or PEX13 to upregulate pexophagy before the activation of mitophagy. We observed that these cells showed impaired mitochondrial clearance compared to control cells (Fig. 3a–f). However, this impairment was not observed in cells depleted of PEX14 (Fig. 3a–f), where pexophagy is not upregulated (Fig. 1b–d). The reduced mitophagy observed with ATG12, PEX1, or PEX13 depletion was not due to defective Parkin recruitment, as Parkin colocalized with mitochondria in OA-treated cells (Fig. 3e) and MFN2, an early substrate of Parkin, was reduced at comparable levels to control cells (Fig. 3a–d). However, the stabilization and colocalization of GFP-Parkin with CVα upon OA treatment in ATG12, PEX1, and PEX13 suggested a defect in the autophagic turnover of mitochondria.

## Increased pexophagy and not the loss of PEX13 impair mitophagy

To confirm that the observed mitophagy impairment was a direct result of upregulated pexophagy and not due to the loss of PEX13 function, we repeated our assay in cells where pexophagy was inactivated. We previously reported that PEX13-dependent pexophagy can be inhibited by co-depleting cells of NBR1[12], an autophagy receptor required for the sequestration of ubiquitinated peroxisomes, but not mitochondria, within autophagosomes[24,27]. Co-depletion of PEX13 and NBR1 rescued the defect in mitochondrial clearance that was observed in cells depleted of PEX13 alone (Fig. 3g, Supplementary Data Fig. 1c, d), supporting that the defect in mitophagy is a result of upregulated pexophagy and not the loss of PEX13 function. Also, rapamycin treatment improved the clearance of mitochondria in cells depleted of PEX1 and PEX13, but not in ATG12-depleted cells, as indicated by CVα fluorescence intensity (Fig. 3h,i), further illustrating that upregulated pexophagy impairs Parkin-dependent mitophagy.

## The absence of peroxisomes augments deferiprone-induced mitophagy

Recently the iron chelator, deferiprone (DFP), was shown to induce mitophagy and pexophagy through a ubiquitin-independent pathway mediated by the autophagy receptor BNIP3L/NIX[28]. Treating cultured cells with DFP upregulated the expression of NIX, a tail-anchored protein that targets both mitochondria and peroxisomes to signal their degradation by autophagy[28]. We took advantage of the co-activation of pexophagy and mitophagy to ask whether pexophagy limits DFP-mediated mitophagy. To address this question, we compared DFP-induced mitophagy in control fibroblasts containing both organelles to DFP-induced mitophagy in peroxisome deficient PEX3-R53ter fibroblast containing only mitochondria[29]. PEX3-R53ter fibroblasts harbor a nonsense mutation in the critical peroxisome biogenesis gene, PEX3[29]. Like ARPE-19, HeLa, and SH-Y5Y cells[28], these fibroblasts express NIX endogenously (Supplementary Data Fig. 4a), however, unlike those cells the fibroblasts display increased sensitivity to DFP as 48 h treatment resulted in cell death. Therefore, we examined mitophagy and pexophagy in the fibroblasts after 28-h of DFP treatment. Following DFP treatment we observed a loss of ATG13 and the conversion of LC3-I:II in both fibroblast lines that indicated autophagy activity (Fig. 4a–c), similar to previous reporting[28]. Further, we observed 60% loss of the peroxisome protein PMP70 in control fibroblasts after DFP treatment (Fig. 4a, d), supporting previous reporting of DFP-induced peroxisome loss[28].

To examine whether mitophagy is influenced by pexophagy upon DFP treatment, we compared different mitochondrial proteins by immunoblot analysis (Fig. 4a). Upon DFP treatment both control and PEX3-R53ter fibroblasts showed a significant loss of MFN2, which was not seen in CCCP-treated fibroblasts (Fig. 4a, e). Given that these fibroblasts do not express Parkin (Supplementary Data Fig. 4b), the lack of mitochondrial loss upon CCCP treatment is not surprising. When compared between the cell lines, there were no significant differences in the basal (DMSO) mitochondrial protein levels between fibroblasts (Fig. 4a, f). However, following DFP treatment we observed a significant increase in the clearance of mitochondrial proteins MFN2 and HSP60 in PEX3-R53ter fibroblasts compared to control (Fig. 4a, e, g).

We confirmed that DFP induced mitophagy in the fibroblasts by examining mitochondrial morphology using the marker HSP60, as well as its colocalization with LAMP1A, a marker for lysosomal organelles (autolysosomes/lysosomes) by confocal microscopy. DFP treatment resulted in mitochondrial fragmentation and clustering near the perinuclear area in both fibroblasts (Fig. 4h, Supplementary Data Fig. 4d). However, the proportion of cells with these mitophagy phenotypes and their severity differed, wherein the PEX3-R53ter fibroblasts displayed increased mitochondrial collapse to the perinuclear area and colocalization with LAMP1A (Fig. 4h, Supplementary Data Fig. 4d). Indeed, quantification of Manders' colocalization coefficient of HSP60 in LAMP1A structures following DFP showed a significant increase compared to untreated cells in PEX3-R53ter fibroblasts, but not in the control fibroblasts (Fig. 4i). Further, we quantified mitochondrial perinuclear localization by cataloging each cell in three different phenotypic catalogs; cells with mitochondria morphology similar to untreated cells were assigned a "mild" phenotype, cells with mitochondria completely collapsed to the perinuclear area were "severe," and cells in between were 'moderate' (Supplementary Data Fig. 4d). We found that PEX3-R52ter cells had a significant increase in the proportion of cells with severe mitochondrial collapse to the perinuclear region than control fibroblasts (Fig. 4j). These findings demonstrate that iron chelation results in a greater degree of mitophagy in the peroxisome deficient PEX3-R53ter fibroblasts compared to control fibroblasts. Taken together with the aggrephagy and Parkin-dependent mitophagy studies, our data suggest that an increased substrate load (pexophagy) can limit other forms of selective autophagy.

## Increased pexophagy impairs the clearance of αS preformed fibrils

The need for cells to tolerate an increased autophagy substrate load is highlighted by diseases where autophagy substrates accumulate, including ZSD that is burdened with perpetual peroxisome degradation. Therefore, we next sought to determine whether disease-

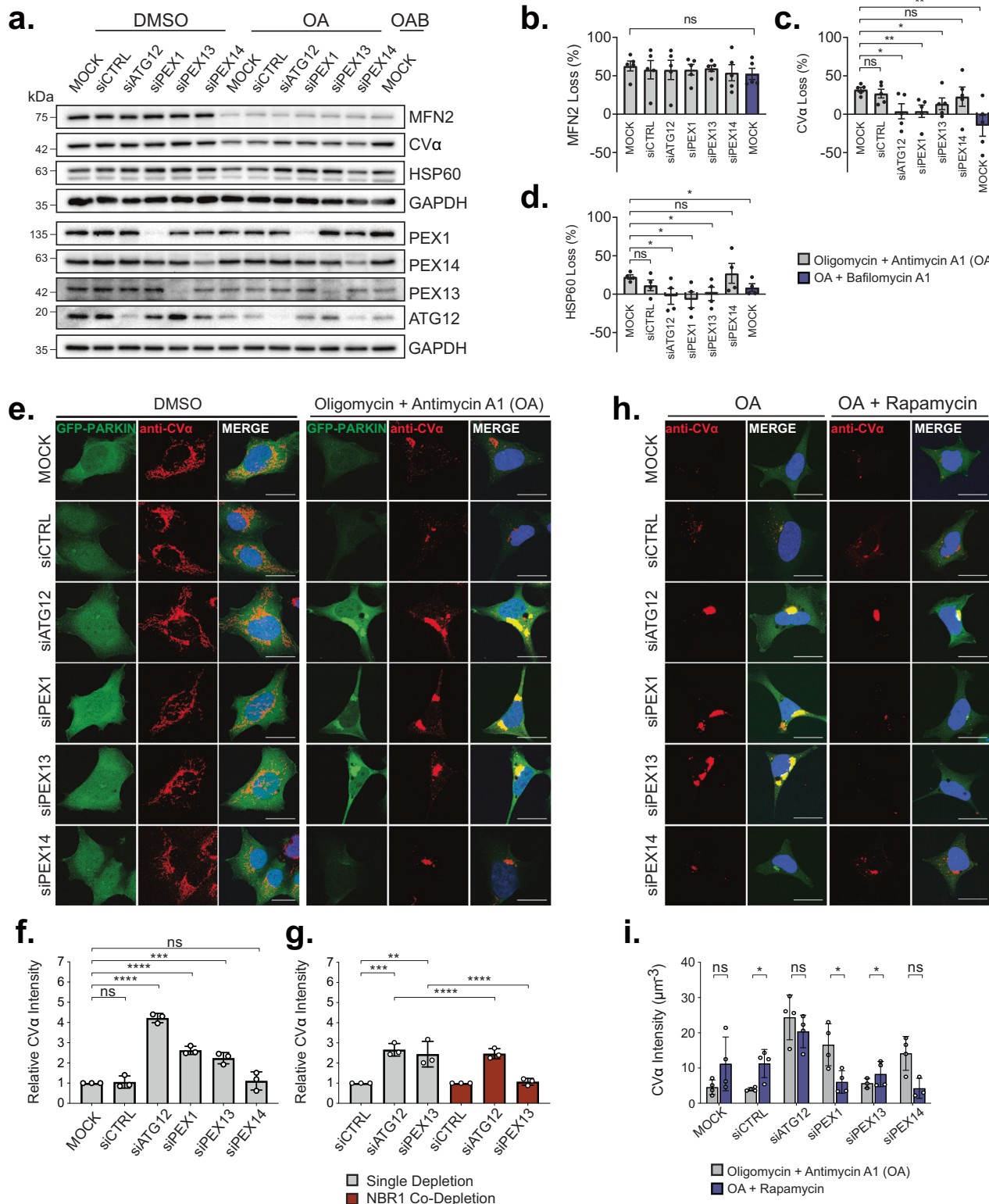

associated substrates of autophagy are affected by upregulated pexophagy. One such substrate is αS, the protein commonly known for its oligomerization and deposition within pathological inclusions in Parkinson's Disease, that has been shown to oligomerize in several ZSD models including PEX13 deficient mice[8]. We examined whether upregulated pexophagy contributes to the accumulation of αS in ZSD by testing whether PEX1 or PEX13 depletion impairs the clearance of αS pre-formed fibrils (PFF) in vitro (Fig. 5a). αS PFF transfected into HeLa cells formed large aggregates within 4-h post transfection and were

cleared within 8-h after washing out the transfection complex (clearance period) (Fig. 5a: α-S PFF vs. α-S + 8 h DMEM). After 8-h, what remained of αS PFF were colocalized with LC3 (Fig. 5a: α-S + 8 h DMEM) suggesting their localization inside autophagosomes. Inhibiting lysosomal protease activity by the addition of leupeptin and E-64 during the clearance period resulted in the persistence of αS aggregates that colocalized with LC3 after 8-h of clearance (Fig. 5a) and the lysosomal marker LAMP1A after 15-h of clearance (Fig. 5b), supporting their autophagic degradation as previously reported[30,31]. Similarly, we found

**Fig. 3 | PEX1 or PEX13 depletion impairs Parkin-dependent mitophagy.**
**a** Immunoblots of GFP-Parkin HEK293 cells subjected to siRNA knockdown prior to treatment for 8-h with DMSO, 2.5 µM Oligomycin and 250 nM Antimycin A1 (OA), or OA + 250 nM Bafilomycin A1. Immunoblots were probed for outer mitochondrial membrane marker MFN2, inner mitochondrial membrane marker CVα, mitochondrial matrix protein HSP60, and siRNA targets: PEX1, PEX13, PEX14, and ATG12.
**b–d** Percentage loss densitometry quantification of the indicated bands normalized to loading control GAPDH. Bands were quantified using ImageJ software.
**e** Representative confocal fluorescent images of GFP-Parkin HEK293 cells treated as in (**a**). Cells immunostaining for CVα as indicated; blue= DAPI. Scale bars, 25 µm.
**f** Quantification of CVα after OA treatment in (**e**) relative to MOCK. CVα was quantified by dividing the total CVα fluorescence intensity by cell volume.

**g** Quantification of CVα after 8-h OA treatment in GFP-Parkin HEK293 cells treated with the indicated siRNA relative to siCTRL conditions (representative images in Fig. S3). **h** Representative images of GFP-Parkin HEK293 cells treated with the indicated siRNA prior to treatment for 8-h with either OA, or OA + 2 µM Rapamycin. Cells were immunostained for mitochondria marker, CVα; blue=DAPI. Scale bars, 25 µm. **i** Quantification of CVα in (**h**). Data are displayed as means from $n = 5$ (**b**, **c**), $n = 4$ (**d**, **i**), $n = 3$ (**f**, **g**) independent experiments ± standard deviation. Statistical significance is denoted as $*P \leq 0.05$, $**P \leq 0.01$, $***P \leq 0.001$, $****P \leq 0.0001$, ns not significant. **b–d** Comparison to MOCK, unpaired $t$-test, two tailed. **f** One-way ANOVA, Dunnett's multiple comparisons test. **g** One-way ANOVA, Tukey's multiple comparisons test. **i** Comparison to OA, unpaired $t$-test, two tailed. Source data and exact $P$ values are provided as a Source Data file.

that cells depleted of ATG12, PEX1, or PEX13 also displayed impaired clearance of αS compared to control or PEX14 depleted cells, manifesting as significantly higher αS fluorescence intensity after an 8-h clearance period (Fig. 5a, c–e). However, unlike the leupeptin/E-64 treated cells, αS did not colocalize with LC3 in these conditions as observed by immunofluorescent imaging (Fig. 5d). Treating cells with rapamycin during the clearance phase improved αS clearance in PEX1 and PEX13 depleted cells with a reduction in αS fluorescence intensity, as well as promoted colocalization of αS with LC3 (Fig. 5d, e). These results indicate that increased pexophagy impairs the clearance of αS by aggrephagy.

### Increased pexophagy impairs aggrephagy in ZSD models
To evaluate whether aggrephagy was also impaired in a ZSD model with increased pexophagy, we repeated the puromycin-induced ALIS assay in PEX1-G843D patient fibroblasts. PEX1-G843D is the most common mutation underlying ZSD, a mutation resulting in peroxisome loss through increased pexophagy (Fig. 6a–c)[13,32,33]. Compared to control fibroblasts, PEX1-G843D cells exhibited a reduction in aggrephagy activity, which was rescued with rapamycin supplementation (Fig. 6d–f). As a control, we also measured aggrephagy activity in the peroxisome-deficient fibroblasts, PEX3-R53ter (Fig. 6a–c)[29]. We observed aggrephagy activity in the PEX3-R53ter fibroblasts similar to control, supporting that the aggrephagy defects in the PEX1-G843D ZSD cells are due to upregulated pexophagy, and not due to a loss of peroxisomes.

### Increased aggrephagy limits pexophagy in proteinopathy models
We next asked whether limiting selective autophagy was a unique property of pexophagy or whether an increase in another selective autophagy pathway similarly lessened the degradation of other substrates. To address this, we examined whether increased aggrephagy would reciprocally influence PEX13-dependent pexophagy. Here we modified our pexophagy-αS PFF aggrephagy assay (Fig. 5) by transfecting HeLa cells with αS 20-h after transfecting cells with siRNA against PEX13 in order to compare the loss of peroxisomes during the early stages of PEX13 knockdown (Fig. 7a). We chose 20-h as the majority of PEX13 was depleted within this time frame (Fig. 7b). Cells were transfected with either αS PFF to induce aggrephagy or with monomeric αS which does not form aggregates as a control[34].

When we transfected αS into HeLa cells 20-h after transfection of PEX13 siRNA, we observed that αS PFF formed large aggregates within 4-h whereas αS monomers did not (Fig. 7c: 0h vs. 4h). Similar to our finding in Fig. 5, αS PFF aggregates resolved in cells treated with control siRNA, whereas the PEX13-depleted cells only cleared about 50% of aggregates after a 15-h clearance period (Fig. 7c, d). To test whether aggrephagy influenced pexophagy, we quantified PMP70-immunostained puncta throughout our assay (Fig. 7c, e). In PEX13-depleted cells transfected with monomeric αS, peroxisomes were significantly reduced by the end of the 39-h assay, supporting upregulated pexophagy (Fig. 7c, e). However, a significant loss of

peroxisomes was not observed in PEX13-depleted cells transfected with αS PFF (Fig. 7c, e), suggesting that increased aggrephagy acutely impairs pexophagy.

To further test whether pexophagy can be inhibited by other forms of selective autophagy, we examined pexophagy in a cell model of Huntington's Disease (HD). HD is an inherited disease caused by an expansion of CAG repeats in the huntingtin gene, which results in the expression of mutant huntingtin (mHTT) with an expanded polyglutamine tract[35]. The exact mechanisms by which mHTT results in HD pathologies are complex, however, it has been shown to form inclusions that are in part removed by aggrephagy[35].

We used a neuronal progenitor cell line established from knock-in mice expressing HTT containing either 7 (STHdh^Q7/7) or 111 (STHdh^Q111/111) glutamine repeats, where the latter have been characterized to express mHTT, and exhibit heightened autophagy compared to STHdh^Q7/7 cells[36,37]. We corroborated these findings by monitoring HTT and LC3 by immunofluorescent staining and immunoblot (Supplementary Data Fig. 5). We observed depleted HTT and LC3-II protein levels and increased colocalization of HTT and LC3 in STHdh^Q111/111 cells compared to STHdh^Q7/7 (Supplementary Data Fig. 5). When treated with BafA1, we observed greater flux of HTT and LC3-II in STHdh^Q111/111 cells compared to STHdh^Q7/7 (Supplementary Data Fig. 5c–h), confirming that STHdh^Q111/111 cells have upregulated aggrephagy.

We induced pexophagy in the HTT cells by ectopically expressing PMP34-GFP-UBKo. We observed no significant difference in basal peroxisome abundance between STHdh^Q111/111 and STHdh^Q7/7 cells (Fig. 8a, b). However, expression of PMP34-GFP-UBKo in STHdh^Q7/7 cells resulted in a significant decrease in peroxisome number (40% loss of PMP70-labeled peroxisomes), whereas no change was observed in STHdh^Q111/111 cells (Fig. 8d–f). These results were supported by immunoblots demonstrating that PMP34-GFP-UBKo expression caused a decrease in PMP70 protein levels in STHdh^Q7/7, but not STHdh^Q111/111 cells, when compared to cells expressing the non-ubiquitin PMP34-GFP construct (Fig. 8g–i). However, inhibiting mTOR with Torin1 for 24-h (Fig. 8c) improved pexophagy activity in STHdh^Q111/111 cells compared to DMSO conditions, as shown by the increased loss of PMP70 punctate structures in immunofluorescent images and PMP70 protein levels (Fig. 8d–i).

Complex autophagic dysfunction has been documented in HD models, and wild-type HTT protein has been proposed to mediate the selective autophagy of protein aggregates, lipid droplets, and mitochondria in response to autophagic stimuli[35,38–40]. Therefore, we investigated whether HTT was required for pexophagy by examining both amino acid starvation and PMP34-GFP-UBKo-induced pexophagy in HeLa cells siRNA-depleted of HTT. In basal culture conditions, treatment with HTT-targeting siRNA achieved a 93% reduction in HTT protein expression but had no effect on peroxisome abundance (Supplementary Data Fig. 6a–f). Further, the HTT-depleted cells showed similar levels of peroxisome loss compared to control when pexophagy was induced by either amino acid starvation (Supplementary Data Fig. 6a–f), or by expressing PMP34-GFP-UBKo (Supplementary Data Fig. 6g–i). Conversely, depletion of ATG12 prevented peroxisome loss following expression of PMP34-GFP-UBKo

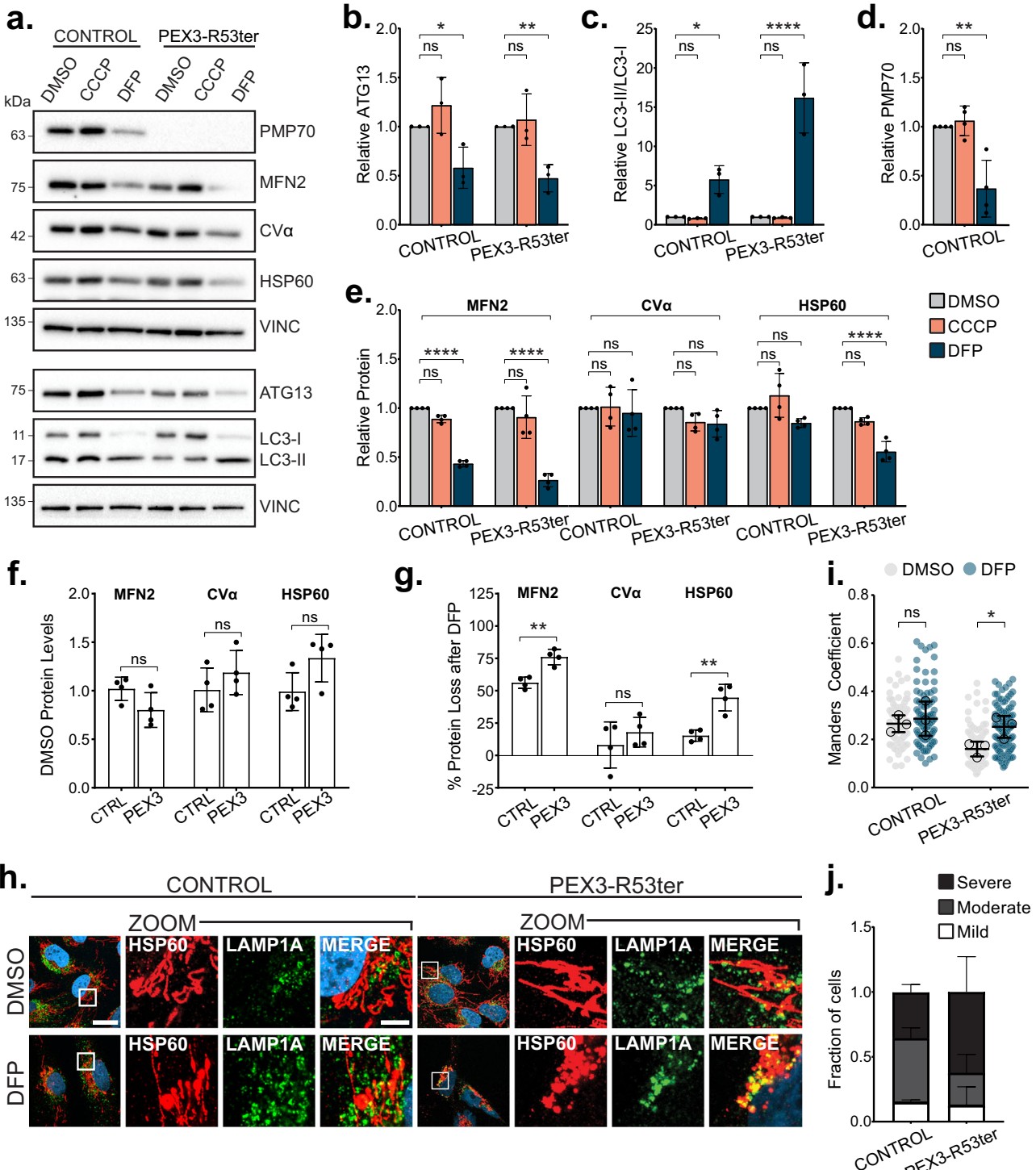

**Fig. 4 | The absence of peroxisomes augments deferiprone-induced mitophagy.**
**a** Immunoblots of control or ZSD fibroblasts treated for 28-h with either DMSO, 10 μM CCCP or 1 mM DFP and probed for the indicated proteins. **b**–**e** Densitometry quantification of the indicated bands from (**a**) normalized to loading control Vinculin and relative to DMSO conditions. **f** Densitometry quantification of the indicated bands from (**a**) in DMSO conditions only, normalized to loading control Vinculin. **g** Percentage loss of band intensity from DMSO to DFP conditions in (**a**), (see Methods). **h** Representative images of control or PEX3-R53Ter ZSD fibroblasts treated for 28-h with either DMSO or 1 mM DFP. Cells were immunostained for HSP60 and LAMP1A; blue=DAPI. Scale bars, 25 μm; zoom scale bars, 5 μm; white

boxes, zoomed region. **i** Manders' correlation coefficient for HSP60 and LAMP1A from (**h**). **j** Quantification of fraction of cells with a mild, moderate, or severe mitophagy phenotype from (**h**). See Methods and Supplementary Data Fig. 4. Data are displayed as individual cells with means overlaid (**i**) or fraction of cells (**j**) from $n = 3$ independent experiments, or means as indicated from $n = 3$ (**b**, **c**) or $n = 4$ (**d**–**g**) independent experiments ± standard deviation. Statistical significance is denoted as *$P \leq 0.05$, **$P \leq 0.01$, ***$P \leq 0.001$, ****$P \leq 0.0001$, ns not significant. **b**–**e** Two-way ANOVA, Dunnett's multiple comparisons test, (**f**, **g**) Unpaired t-test, two tailed, **i** Two-way ANOVA, Tukey's multiple comparisons test using means. Source data and exact P values are provided as a Source Data file.

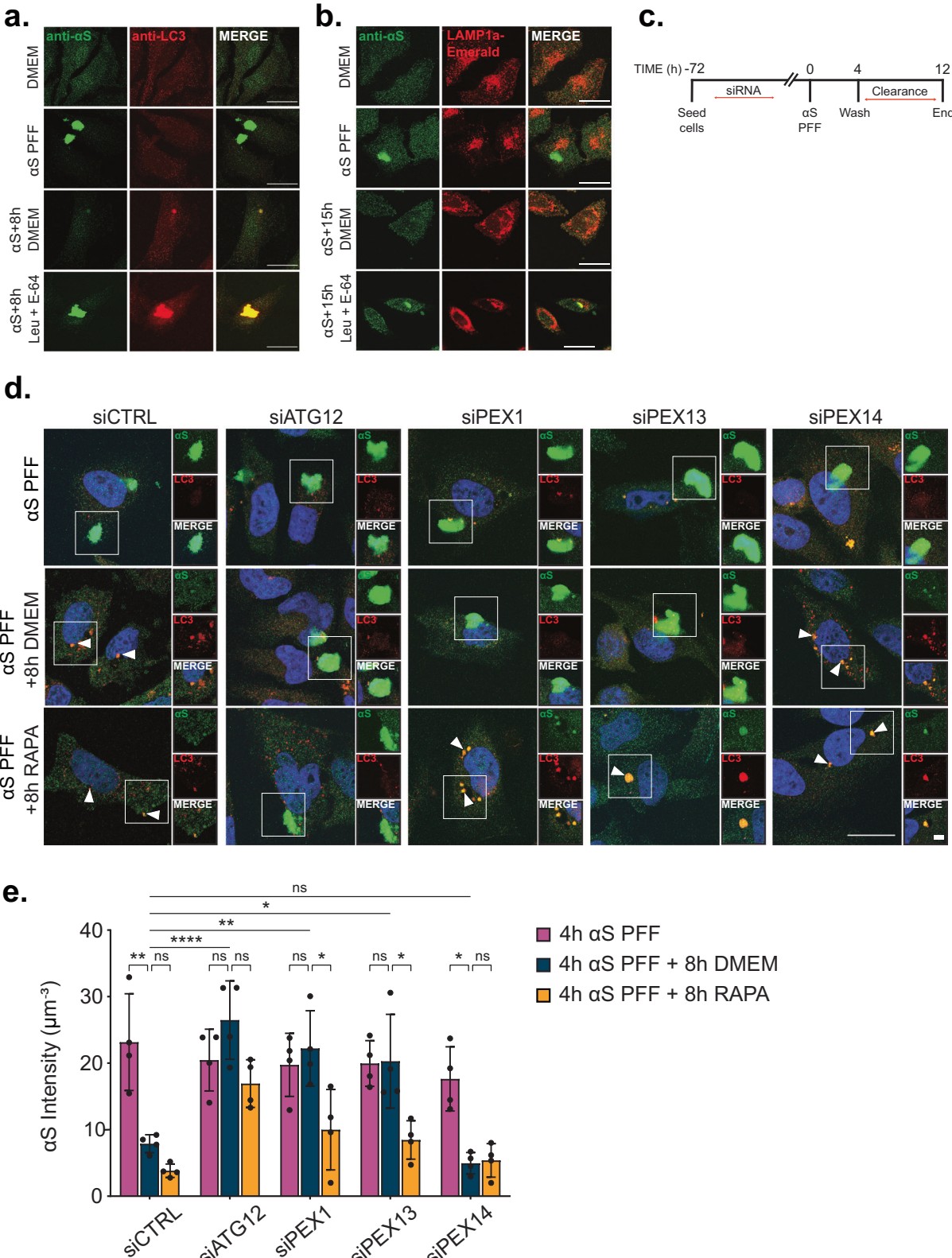

(Supplementary Data Fig. 6g–i). These findings suggest that the loss of functional HTT protein does not impair pexophagy but supports a model wherein STHdh^Q111/111 cells with upregulated aggrephagy have decreased capacity for pexophagy compared to STHdh^Q7/7 cells, which can be improved by upregulating autophagy.

Altogether, our findings in proteinopathic αS and mHTT cell models suggest that limiting selective autophagy induced by an upregulation of one form of selective autophagy is not unique to pexophagy.

**Autophagy receptors are not uniformly limited during upregulated pexophagy**

To understand the mechanism by which pexophagy impairs aggrephagy and mitophagy, we sought to determine the rate-limiting steps/

**Fig. 5 | PEX1 or PEX13 depletion impairs the autophagic clearance of αS pre-formed fibrils. a** HeLa cells at each stage in the αS assay: untreated (DMEM), 4-h transfection with αS pre-formed fibrils (PFF); 4-h αS PFF followed by an 8-h clearance period in either DMEM or 0.25 mM Leupeptin and 2 μM E-64. Cells were immunostained for αS and LC3. Scale bars, 25 μm. **b** HeLa cells immunostained for αS and expressing LAMP1a-mCherry at each stage in the assay as in (**a**), but here with a 15-h clearance period. Scale bars, 25 μm. **c** Schematic of assay where cells are treated with siRNA before αS PFF transfection. **d** Representative images from HeLa cells treated with the indicated siRNA and subjected to the αS assay in (**c**), given an

8-h clearance period in either DMEM or 2 μM Rapamycin. Cells were immunostained for αS and LC3; blue = DAPI. Scale bars, 25 μm; cropped image scale bars, 5 μm; white arrows indicate colocalization of αS and LC3. **e** Quantification of αS intensity in (**d**), calculated by dividing the total αS florescence intensity by cell volume. Data are displayed as means ± standard deviation from *n* = 4 independent experiments. Statistical significance is denoted as *$P \le 0.05$, **$P \le 0.01$, ***$P \le 0.001$, ****$P \le 0.0001$, ns not significant. Two-way ANOVA, Tukey's multiple comparisons test. Images in (**a**) and (**b**) are representative of *n* = 3 independent experiments. Source data and exact *P* values are provided as a Source Data file.

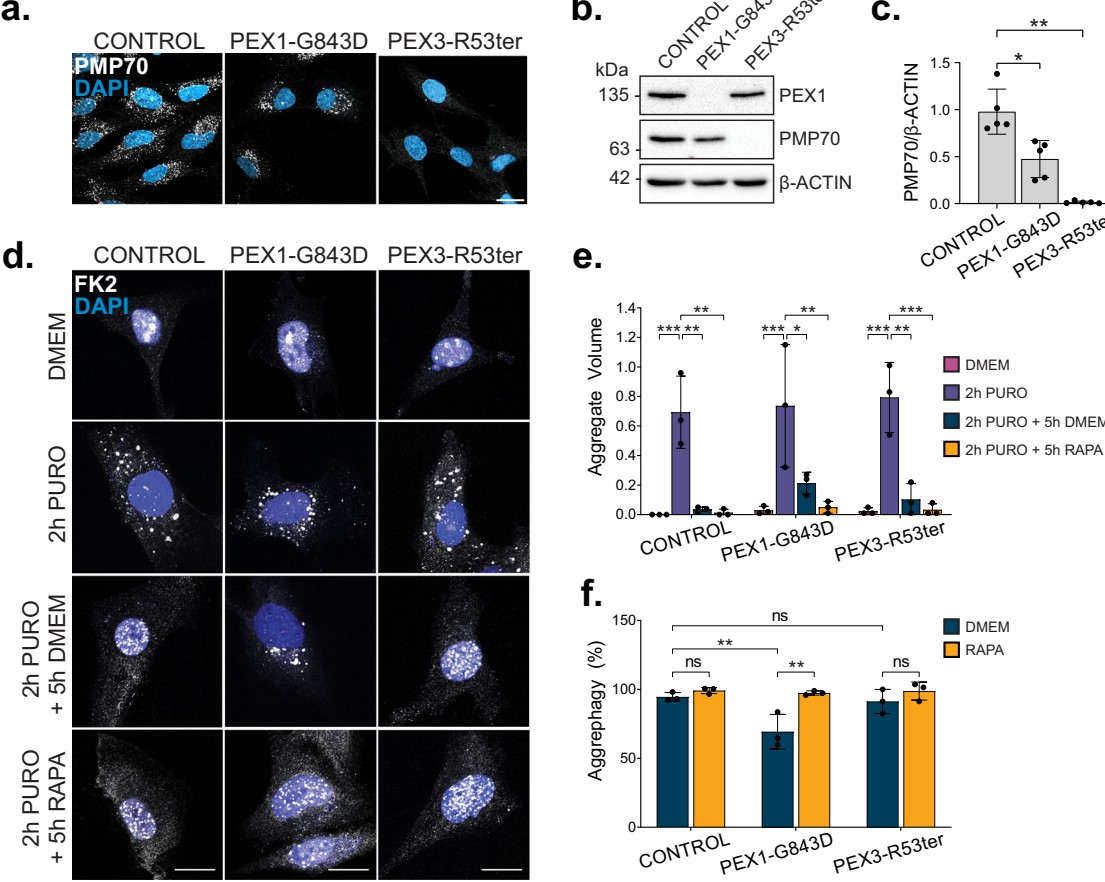

**Fig. 6 | PEX1-G843D Zellweger Spectrum Disorder patient fibroblasts exhibit impaired aggrephagy. a** Images of control, PEX1-G843D, or PEX3-R53Ter ZSD fibroblasts stained for PMP70; blue = DAPI. Scale bars, 25 μm. **b** Immunoblot of ZSD fibroblasts probed for PEX1, PMP70, and β-Actin. **c** Densitometry quantification of PMP70 bands in (**b**) normalized to β-Actin. **d** Representative images of ZSD fibroblasts at each stage in the aggrephagy assay. Cells were immunostained with the antibody FK2; blue=DAPI. Scale bars, 25 μm. **e** Quantification of Aggregate Volume in (**a**). Aggregate Volume was calculated by dividing the total volume of FK2 puncta

by cell volume. **f** Quantification of aggrephagy activity in (**d**), relative to control fibroblasts. Aggrephagy activity was measured as the % clearance of FK2 aggregates. Data are displayed as means ± standard deviation from *n* = 5 (**c**) or *n* = 3 (**e**, **f**) independent experiments. Statistical significance is denoted as *$P \le 0.05$, **$P \le 0.01$, ***$P \le 0.001$, ****$P \le 0.0001$, ns not significant. **c** One-way ANOVA, Dunnett's multiple comparisons test. **e**, **f** Two-way ANOVA, Tukey's multiple comparisons test. Source data and exact *P* values are provided as a Source Data file.

components during conditions of upregulated pexophagy. Autophagy receptors are essential for selective autophagy and can be specific to a single substrate or exhibit redundancy between multiple substrates. Both protein aggregates and peroxisomes are targeted for autophagy by the receptors NBR1 and p62, whereas ubiquitin-dependent mitophagy is primarily mediated by OPTN and NDP52[24,27,41]. Therefore, we assessed whether increased pexophagy delays the degradation of other substrates by consuming autophagy receptors by immunoblot analysis of autophagy receptors. In cells depleted of ATG12, we observed increased abundance of all autophagy receptors compared to mock-transfected cells, confirming impaired autophagic flux (Supplementary Data Fig. 7a, b). In cells depleted of either PEX1 or PEX13, we observed that OPTN abundance was increased to similar levels as seen with ATG12 depletion,

supporting a defect in mitophagy flux (Supplementary Data Fig. 7a, b). As previously shown, the depletion of PEX1 resulted in decreased p62 and NBR1 abundance, which could suggest that these receptors become limiting during PEX1 depletion (Supplementary Data Fig. 7a, b)[13]. However, PEX13 depletion caused no change to NBR1 abundance and an increase in p62 abundance (Supplementary Data Fig. 7a, b). Since cells depleted of PEX14 exhibited a similar increase in p62 as those depleted of PEX13, the upregulation of p62 may not be related to pexophagy but to other functions shared by PEX13 and PEX14 which together form the peroxisome import pore[42]. Taken together, these findings demonstrate that autophagy receptor abundance is affected in PEX1 or PEX13 depleted cells, however their changes do not uniformly support that autophagy receptors are rate-limiting for selective autophagy.

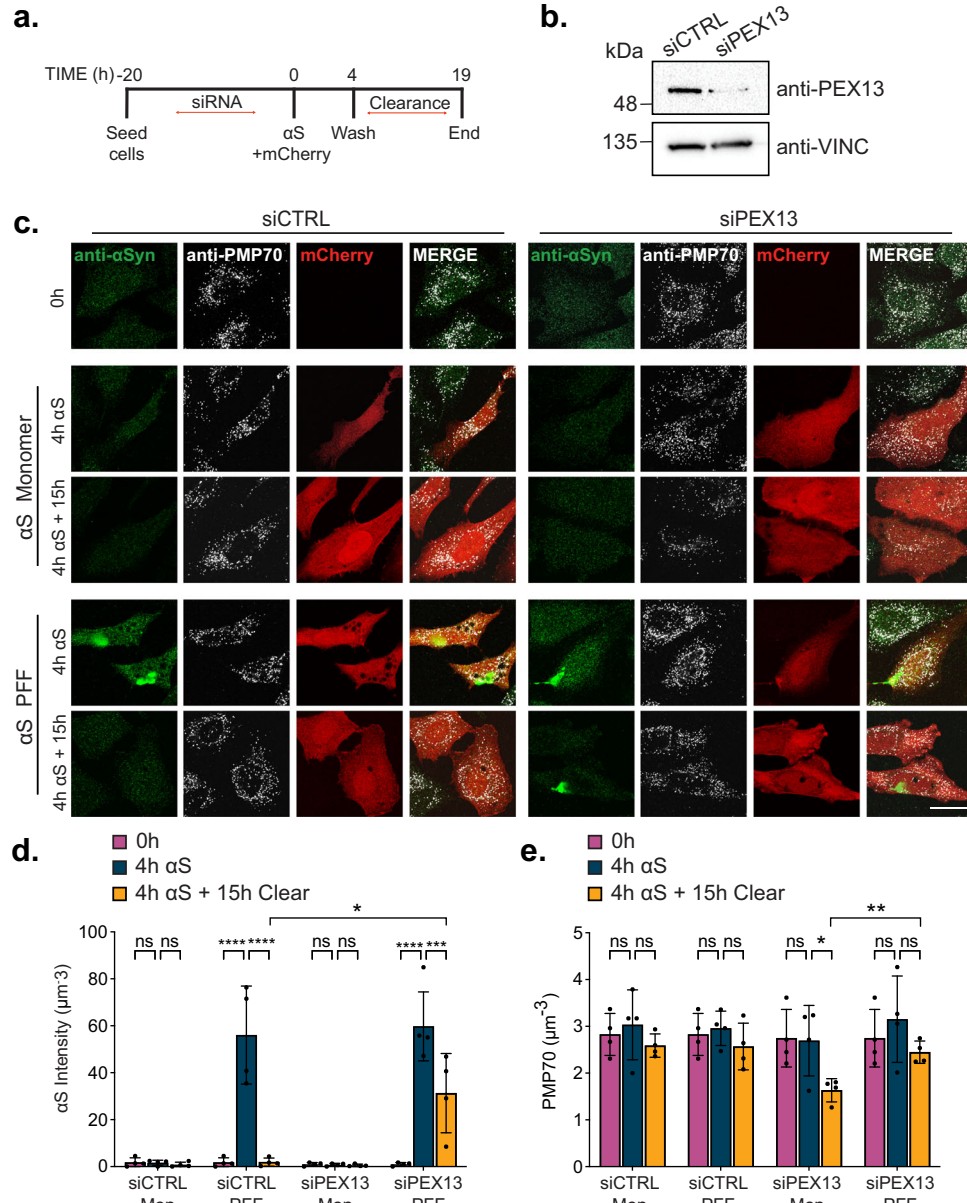

**Fig. 7 | αS PFF degradation limits pexophagy induced by PEX13 depletion.**
**a** Schematic of assay. 20-h after a single siRNA transfection, αS monomers (Mon) or PFF are co-transfected into cells with an mCherry plasmid for 4-h, followed by a 15-h clearance period in DMEM. **b** Immunoblot of HeLa cells treated with the indicated siRNA for 20-h and probed for PEX13 and β-Actin; representation of n = 3 independent experiments. **c** Representative images of HeLa cells treated with the indicated siRNA and transfected with either αS Mon or PFF at each stage in the assay, and immunostained for PMP70. Scale bars, 25 μm. **d** Quantification of αS intensity in (**c**), calculated by dividing the total αS florescence intensity by cell volume. **e** Quantification of PMP70 in (**c**). PMP70 was calculated by dividing the number of PMP70 puncta by cell volume. Data are displayed as means ± standard deviation from n = 4 independent experiments. Statistical significance is denoted as *$P \leq 0.05$, **$P \leq 0.01$, ***$P \leq 0.001$, ****$P \leq 0.0001$, ns not significant. Two-way ANOVA, Tukey's multiple comparisons test. Comparison of siCTRL to siPEX13 (**d**) and siPEX13 Mon to PFF (**e**), unpaired t-test, two tailed. Source data and exact P values are provided as a Source Data file.

## Autophagosome formation is not suppressed by upregulated pexophagy

The rescue of aggrephagy and mitophagy by rapamycin in PEX1 and PEX13 depleted cells (Figs. 1g and 3d) suggests that an mTORC1-regulated step(s) of autophagy is limited in cells with upregulated pexophagy. Therefore, we next asked whether autophagosome formation or flux may be suppressed in PEX1 or PEX13 depleted cells. We measured basal autophagy flux by treating cells with BafA1 and monitoring the LC3-II:I ratio by immunoblot. A significant increase in LC3-II:I ratio following BafA1 treatment compared to DMSO treatment was only observed in PEX1 or PEX13 depleted cells (Supplementary Data Fig. 7c, d), supporting an increase in pexophagy in these cells.

Rapamycin treatment significantly increased the LC3-II:I ratio in all conditions except ATG12 depletion, demonstrating that PEX1 and PEX13 depleted cells have the capacity to further upregulate autophagosome biogenesis (Supplementary Data Fig. 7c–f).

We confirmed these findings by monitoring the abundance of LC3 puncta by immunofluorescence in DMSO or rapamycin-treated cells. Cells were transfected with Cherry-LC3 and permeabilized with digitonin prior to fixation in order to wash out cytosolic LC3-I, such that only membrane-bound LC3-II remained (Supplementary Data Fig. 8g). Here, we observed that PEX1 or PEX13 depleted cells had increased LC3-II puncta compared to mock-transfected cells, but this difference was ablated by rapamycin (Supplementary Data Fig. 8g, h), supporting

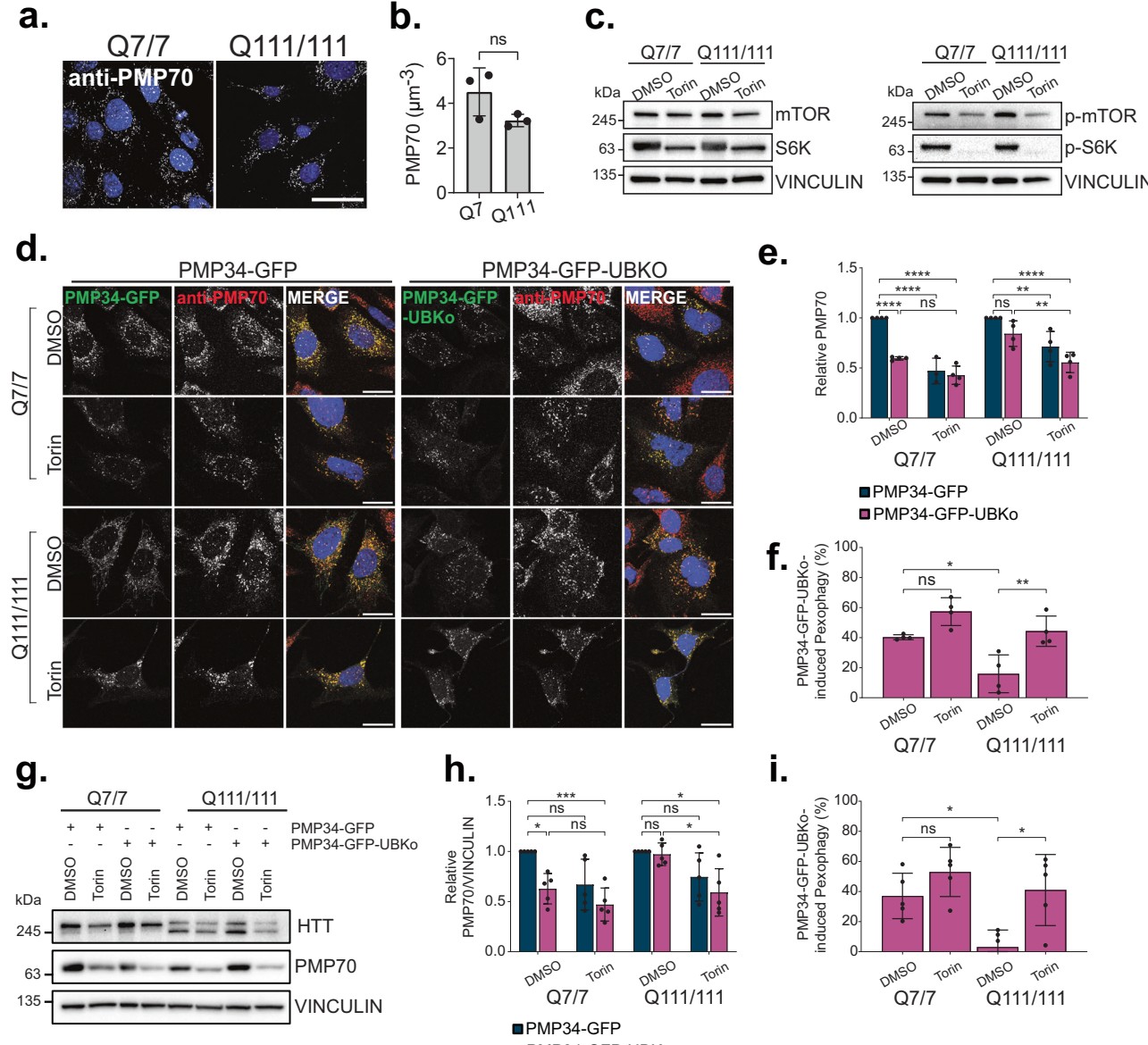

**Fig. 8 | STHdh^(QIII/III) cells exhibit impaired pexophagy. a** Immunofluorescent images of STHdh^(Q7/7) or STHdh^(QIII/III) cells stained for peroxisomal marker, PMP70; blue=DAPI. Scale bar, 25 μm. **b** Quantification of PMP70 in (**a**). PMP70 was calculated by dividing the number of PMP70 puncta by cell volume. **c** Immunoblots of STHdh^(Q7/7) or STHdh^(QIII/III) cells treated for 24-h with 100 nM Torin1 and probed for mTOR, S6K, p-mTOR, p-S6K, and loading control Vinculin; representative of n = 3 independent experiments. **d** Representative images of STHdh^(Q7/7) or STHdh^(QIII/III) cells expressing either PMP34-GFP or PMP34-GFP-UBKo for 48-h, and immunostained for PMP70; blue = DAPI. Cells were grown in either DMSO or 100 nM Torin1 for the final 24-h. Scale bars, 25 μm. **e** Quantification of PMP70 in (**d**), relative to DMSO PMP34-GFP condition for each cell type. **f** Quantification of PMP34-GFP-

UBKo-induced pexophagy in (**d**). Pexophagy was calculated as the percentage loss of PMP70 from DMSO PMP34-GFP conditions (see Methods). **g** Immunoblots of STHdh^(Q7/7) or STHdh^(QIII/III) cells treated as in (**d**), probed for HTT, PMP70, and loading control Vinculin. **h** Densitometry quantifications of PMP70 bands normalized to Vinculin and relative to DMSO PMP34-GFP condition for each cell type. **i** Quantification of PMP34-GFP-UBKo-induced pexophagy in (**g**). Data are displayed as means ± standard deviation from n = 3 (**b**), n = 4 (**e, f**), or n = 5 (**h, i**) independent experiments. Statistical significance is denoted as *$P ≤ 0.05$, **$P ≤ 0.01$, ***$P ≤ 0.001$, ****$P ≤ 0.0001$, ns not significant. **b** Unpaired $t$-test, two tailed. **e, h** Two-way ANOVA, Tukey's multiple comparisons test. **f, i** One-way ANOVA, Tukey's multiple comparisons test. Source data and exact $P$ values are provided as a Source Data file.

increased basal autophagosome formation in the pexophagy upregulated cells. Together, these findings suggest that PEX1 and PEX13 depletion does not inhibit autophagosome formation or flux, but instead shows increased autophagosome formation.

## ULK1 rescues limited selective autophagy

Our LC3 studies suggest that upregulated pexophagy does not limit selective autophagy of other substrates by impairing autophagosome formation. In contrast, we observed a moderate increase in autophagosome in cells depleted of PEX1 or PEX13 compared to the controls

(Supplementary Data Fig. 7c–h), even though the mTORC1 activation state was not different between cells (Supplementary Data Fig. 3). Similarly, we did not find any difference in the mTORC1 mediated phosphorylation state of ULK1, the regulatory subunit of the ULK1 complex, which is essential for the initiation of autophagosome formation[43,44] (Supplementary Data Fig. 8a,b). However, there is some evidence that ULK1 initiates the biogenesis of sequestering membranes for mitophagy independent of inhibitory mTORC1 activity[45]. Therefore, we asked whether ULK1 may become limited in cells with upregulated pexophagy.

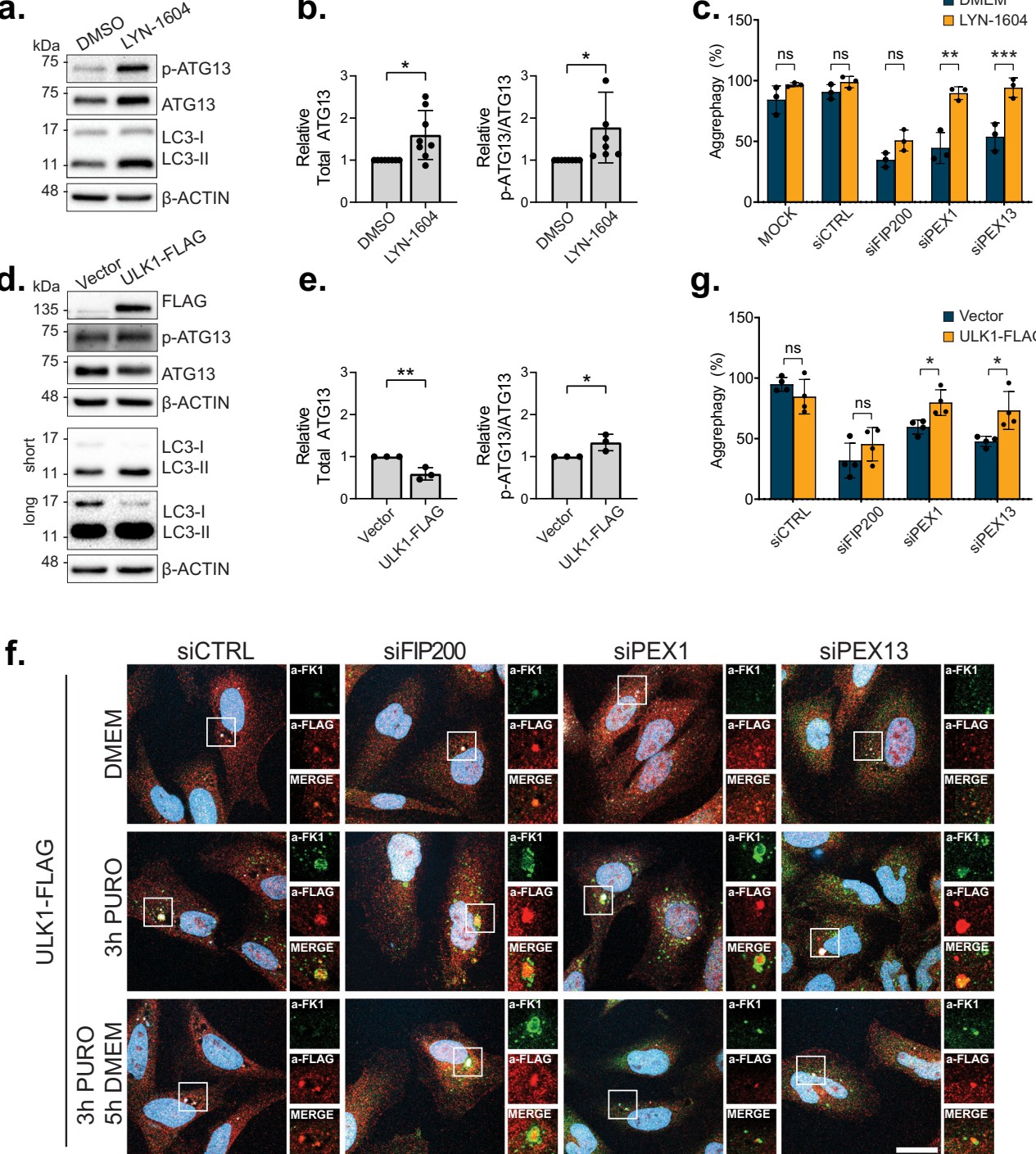

**Fig. 9 | Impaired aggrephagy during PEX1 or PEX13 depletion can be rescued by ULK1. a** Immunoblot of HeLa cells treated for 5-h with either DMSO or 20 μM LYN-1604 and probed for the indicated proteins. **b** Densitometry quantifications of ATG13 bands normalized to β-Actin, or the ratio of p-ATG13 band intensity/ATG13 band intensity from (**a**). **c** Quantification of aggrephagy activity in cells treated with the indicated siRNA and subjected to the puromycin-aggrephagy assay with a clearance period in either DMEM or LYN-1604 (representative images in Supplementary Data Fig. 8c). Aggrephagy activity was measured as the % clearance of FK1 aggregates, relative to mock transfected "3 h P + 5 h DMEM" cells. **d** Immunoblot of HeLa cells electroporated with an empty vector control or ULK1-FLAG and probed for the indicated proteins. **e** Densitometry quantifications of ATG13 bands normalized to β-Actin, or the ratio of p-ATG13 band intensity/ATG13 band intensity

from (**d**). **f** HeLa cells treated with the indicated siRNA prior to transfection of an empty vector (Supplementary Data Fig. 8f) or ULK1-FLAG and subjection to the puromycin-aggrephagy assay. Representative images of cells at each stage in the assay: DMEM, 3-h 5 μg mL$^{-1}$ Puromycin, or 3-h 5 μg mL$^{-1}$ Puromycin followed by 5-h clearance period in DMEM. Cells were immunostained for FK2 and FLAG; blue=-DAPI. Scale bars, 25 μm; white boxes, zoomed region. **g** Quantification of aggrephagy activity of cells in (**f** Supplementary Data Fig. 8f), relative to empty vector transfected "3 h P + 5 h DMEM" cells. Data are displayed as means from $n = 6$ (**b**), $n = 3$ (**c,e**), or $n = 4$ (**g**) independent trials ± standard deviation. Statistical significance is denoted as *$P \leq 0.05$, **$P \leq 0.01$, ***$P \leq 0.001$, ****$P \leq 0.0001$, ns not significant. Unpaired $t$-test, two tailed. Source data and exact $P$ values are provided as a Source Data file.

To test whether ULK1 is a limiting component during upregulated pexophagy, we first increased activated ULK1 using the ULK1/2 agonist, LYN-1604, during the clearance period of our puromycin-aggrephagy assay. Supplementation of LYN-1604 for 5-h resulted in a robust increase in ULK1 activity, evidenced by increased stability and phosphorylation of its well-established substrate, ATG13 (S318), in addition to an increased LC3-II:I ratio, as previously described (Fig. 9a, b)[46]. LYN-1604 significantly improved aggrephagy activity in cells depleted of PEX1 or PEX13, but not in cells depleted of the ULK1 complex protein FIP200 (Fig. 9c and Supplementary Data Fig. 8c–e). Next, we repeated our puromycin-aggrephagy assay in cells overexpressing FLAG tagged ULK1. Like LYN-1604, ULK1-FLAG overexpression led to a modest increase in ATG13 phosphorylation and increased the LC3-II:I ratio compared to a vector control (Fig. 9d, e), suggesting that exogenously expressing ULK1 can increase autophagy. Similar to activation of ULK1 by LYN-1604, the overexpression of ULK1-FLAG rescued the impaired aggrephagy activity in cells depleted of PEX1 or PEX13 (Fig. 9f, g and Supplementary Data Fig. 8f, g). Collectively, these data suggest that ULK1 is a limiting component during upregulated pexophagy.

## Discussion

A hallmark of proteinopathies such as neurodegenerative[47–49] and liver[50,51] disease is the accumulation of damaged organelles like mitochondria and ER, which correlates with the buildup of misfolded protein. Despite the presence of quality control processes such as autophagy, it is unclear why these harmful materials are not removed. There are two popular hypotheses for this phenomenon. One is that autophagy activity decreases with age[52–54], leading to a greater buildup of protein aggregates and damaged organelles. The other is that pathogenic proteins associated with proteinopathies can directly inhibit the autophagic degradation of cellular components. Here, we provide evidence for a third model, where the capacity of selective autophagy in a given cell is limited, such that a pathogenic increase in one substrate of selective autophagy hinders the degradation of other components.

Our work has identified limited selective autophagy as a possible contributor to the accumulation of damaged organelles in ZSD. ZSD is not a proteinopathy, but the defect in peroxisomal genes often results in the accumulation of damaged mitochondria in diverse tissues and prefibrillar aggregates of αS in neurons[8–10]. However, why defective mitochondria accumulate instead of being eliminated by autophagy is not known. In this study, we show that an increase in pexophagy activity limits the capacity of the cell to clear damaged mitochondria and protein aggregates. Cellular depletion of PEX13 or PEX1 (the most commonly mutated gene in ZSD) does not prevent the formation of peroxisome membrane structures but reduces their number by constantly degrading new peroxisomes resulting in upregulated pexophagy (Fig. 1)[12,13]. This upregulation of pexophagy reduced the autophagic clearance of ALIS (Fig. 1), damaged mitochondria (Fig. 3), and αS (Fig. 5). The accumulation of these cytotoxins was not directly caused by the loss of PEX1 or PEX13 function, or the loss of other peroxisome functions, as cells absent of peroxisomes due to a defect in PEX3 or PEX19 showed no impairments in aggrephagy or mitophagy. Instead, upregulated cellular pexophagy activity was responsible for the reduction in mitophagy and aggrephagy activity, as inhibiting pexophagy rescued the phenotype (Fig. 1h–j, Fig. 3g, and Supplementary Data Fig. 1c, d).

A reduction in selective autophagy capacity of a cell due to pexophagy may also contribute to the accumulation of autophagic substrates in ZSD resulting from other peroxin mutations. Both damaged mitochondria and αS oligomers have been observed in PEX2 and PEX5 knockout mice[8,10], yet the loss of these peroxins does not directly induce pexophagy. Both PEX2 and PEX5 are involved in designating peroxisomes for pexophagy, as PEX2 ubiquitinates peroxisomal proteins and PEX5 is the most commonly ubiquitinated peroxisomal

protein that signals pexophagy[55]. A possible explanation may reside in the non-functional 'ghost peroxisomes' formed in the absence of PEX2 and PEX5 that are readily targeted for autophagic degradation in various yeast models[56,57]. Although the fate of mammalian ghost peroxisomes has not been studied, the E3 ubiquitin ligase STUB1 was recently shown to target oxidatively damaged peroxisomes for degradation by pexophagy[58], suggesting that damaged or incomplete peroxisomes are likely degraded by autophagy.

Taken together, we propose that in ZSD cases associated with increased pexophagy, the loss of peroxisomes has multifactorial consequences on cellular health. First, the loss of peroxisome function bears a host of insults, including peroxin mislocalization[59], increased oxidative stress[9,10], and a reduction in fatty acid oxidation[8,10] that contribute to mitochondrial damage and protein aggregation. In parallel, the sustained increase in pexophagy utilizes the limited selective autophagy capacity of the cell, thereby reducing the clearance of other substrates and perpetuating their accumulation. Damaged mitochondria likely contribute to the overall substrate load to further stress the selective autophagy pathways, as is observed in neurodegenerative disease[60]. As such, in ZSD cases associated with pexophagy-independent mutations, an accumulation of damaged mitochondria due to the loss of peroxisome function may similarly limit cellular selective autophagy pathways.

In our pexophagy system, we identified ULK1 as a limiting component of autophagy. We found that overexpressing ULK1 or chemically activating it with LYN-1604 rescued selective autophagy of aggregates, suggesting that activated ULK1 limits selective autophagy induction in cells with elevated pexophagy. However, we cannot rule out that other components of sequestering membrane biogenesis machinery may also be limited, including subunits of the ULK1 complex, the VPS34 phosphatidylinositol 3 kinase complex, or the TANK-binding kinase 1 (TBK1). As the ULK1 complex is recruited to autophagy substrates likely together with TBK1 to drive the recruitment of VPS34[45,61,62], it is possible that these complexes are collectively limiting due to their dual localization and stabilizing interactions. The autophagy receptor NIX is likely a limited component in DFP-induced mitophagy/pexophagy, however, it is unlikely that ubiquitin-dependent autophagy receptor abundance in our pexophagy-induced systems is restricting selective autophagy. Although some autophagy receptors were decreased in cells with upregulated pexophagy, others were increased, and there was little commonality between PEX1 and PEX13 depleted cells, suggesting that autophagy receptor abundance was not uniformly limited, whereas their oligomeric or posttranslational states remain to be investigated.

It is unclear why cells with an increased substrate load do not upregulate activated ULK1 in our assays. We observed a moderate increase in autophagosomes in both PEX1 and PEX13 depleted cells which we attribute to increased pexophagy, yet the phosphorylation status of mTORC1 and ULK1 is unchanged between cells with upregulated pexophagy and controls, suggesting that their autophagy capacity is similar. The conformation, localization, and kinase activity of ULK1 is post-translationally regulated, and during energy abundance conditions ULK1 activity is largely inhibited to limit autophagosomes, suggesting that upregulated pexophagy may consume the pool of ULK1 that is competent for selective autophagy. Thus, inhibiting mTORC1 allows the rescue of aggrephagy and mitophagy. As LYN-1604 increases the global phosphorylation of ULK1 at S317 and decreases phosphorylation at S757[46], this suggests that ULK1 species dephosphorylated at S757 may limit selective autophagy during increased pexophagy. However, this does not rule out that other ULK1 species may be competent for selective autophagy. Recent work by three different groups demonstrated that the autophagy receptors p62, TAX1BP1, and NDP52 can recruit the ULK1 complex directly to substrates to drive autophagosome formation locally[45,61,63]. At least for the NDP52-dependent recruitment of ULK1 to mitochondria, the

activation of ULK1 was independent of either mTORC1 and AMPK phosphorylation[45]. A possible mechanism for ULK1 activation may be the autophosphorylation of ULK1 induced by its clustering at substrates. During energy-replete conditions in *S. cerevisiae*, the clustering of the yeast homolog for ULK1, Atg1, was sufficient to induce autophosphorylation and increase its kinase activity[64,65]. Therefore, oligomerized and autophosphorylated ULK1 may represent an alternate species of ULK1 that could limit selective autophagy, which could explain why overexpression of ULK1 is sufficient to rescue aggrephagy (Fig. 9). Further, it is also possible that alternate ULK1 species exist, which are post-translationally modified at sites other than the AMPK1 or mTORC1 phosphorylated residues.

A previous study reported that the loss of PEX13 results in impaired autophagic clearance of mitochondria and Sindbis virus, and proposed that PEX13 plays a direct role in mitophagy and virophagy[11]. We have recently identified that PEX13 is required to prevent pexophagy of healthy peroxisomes and is degraded to mediate pexophagy during starvation[12]. Here, we show that loss of PEX13 alone is not responsible for preventing mitophagy or aggrephagy, as artificially targeting a ubiquitin motif to peroxisomes to induce pexophagy was sufficient to impair aggrephagy. The role of PEX13 in pexophagy was likely missed by ref. 11 due to the short depletion time frame (2-day siRNA treatment), as we found that significant loss of peroxisomes was observed 3 days post-siRNA treatment (Fig. 1).

Using ZSD models, we provide here evidence to support that distinct selective autophagy pathways can influence each other, and that the degradative capacity of selective autophagy can be acutely limited by an increased substrate load in cells. Our studies provide justification for targeting the activation of selective autophagy or inhibition of pexophagy as a strategy for ZSD treatment. However, such studies must weight how such activation/inhibition can affect different tissues and cell types. Autophagy is highly regulated, and it is not known how long-term activation of autophagy affects neurons or glia. Similarly, it is not clear what effect an accumulation of damaged or partially assembled peroxisomes has on the regulation of both lipid and redox homeostasis, which could impact autophagy, therefore future studies of pexophagy in ZSD model systems are required. Further, it remains to be determined whether a similar mechanism of limited autophagy capacity caused by an increase in one specific substrate contributes to the cellular pathophysiology of other diseases, including proteinopathy and mitochondrial disease.

## Methods
The reagents used in this study including chemicals and commercially available kits, are summarized in Supplementary Table 1.

### Constructs and siRNA
The PMP34-GFP[66] and PMP34-GFP-UBko[19] constructs used in this study were previously generated using standard protocols, where the PCR product of the ORF of PMP34 was ligated into the *Eco*RI and *Bam*HI site of the pmGFP-N1 vector (Clontech). Ub-KO (G76V), a ubiquitin where all seven lysines were mutated to arginines, was excised from GFP-Ub-KO (G76V) (Addgene 11932) with *Bsr*GI and *Not*I, and the resulting fragments were ligated into similar sites in PMP34-GFP. The Cherry-LC3 construct was previously generated[27], where the PCR product of the LC3-B ORF was ligated into the *Bgl*II-*Eco*RI site of the mCherry-C1 vector (Clontech) using standard protocols. LAMP1A-mEmerald was donated by Dr. Sergio Grinstein (Hospital for Sick Children, Toronto, ON, Canada), and FLAG-ULK1-wild type was donated by Dr. John Brumell (Hospital for Sick Children, Toronto, ON, Canada). The siRNAs used in this study were custom synthesized from Sigma-Aldrich and are listed in Supplementary Table 2. siRNA knockdown was validated by immunoblot or TaqMan real-time quantitative PCR.

### Cell culture and transfection
HeLa cells were purchased from American Type Culture Collection (CCL-2).

Stably transfected GFP-Parkin HEK293 cells were a gift from Dr. Angus McQuibban (University of Toronto, Toronto, ON, Canada). The immortalized skin fibroblast cell lines from healthy (Control), PEX1-G843D, and PEX3-R53ter patients[13] were gifts from Dr. Nancy Braverman (McGill University, Montreal, QC, Canada), while the PEX19-deficient (PBD399-T1)[20] immortalized human fibroblast cell line was a gift from S.J Gould (John Hopkins University School of Medicine, Baltimore, MD). STHdh$^{Q111/111}$ and STHdh$^{Q7/7}$ cells were donated by Dr. Ray Truant (McMaster University, Hamilton, ON, Canada), and are previously described[36]. All cells were cultured in Dulbecco's modified Eagle's medium (Gibco) supplemented with 10% fetal bovine serum (Wisent), at 5% $CO_2$ in a 37 °C humidified incubator. Cells routinely tested negative for mycoplasma (FroggaBio, 25235). For amino acid starvation experiments, cells were washed twice with PBS and subjected to HBSS for the indicated time (Lonza). siRNA and plasmid transfections were performed using Lipofectamine 2000 (Invitrogen) according to the manufacturer's instructions. For all knockdown experiments, a 2-day siRNA transfection was performed followed by a 24-h recovery period before chemical treatments were applied. The exception is Fig. 7, where a single siRNA transfection was performed followed by a 20-h recovery period prior to αS transfection. In experiments where cells were transfected with plasmids, plasmids were expressed for 24-48-h preceding chemical treatments, as specified in the figure captions. For experiments performed in STHdh$^{Q111/111}$ and STHdh$^{Q7/7}$ cells as well as the ULK1-FLAG experiments performed in HeLa cells, transfection was performed using the Neon transfection system (Invitrogen) according to the manufacturer's instructions.

### Immunoblotting
Cells were washed with PBS and lysed in 100 mM Tris, pH 9, 1% SDS with 1X protease cocktail inhibitor (BioShop, PIC002.1). For analysis of phosphorylated proteins, lysis buffer was supplemented with 20 mM NaF and 5 mM $Na_2VO_4$. Lysates were boiled for 15-min at 95 °C with vortexing every 5-min, and then centrifuged at 13,000 $g$ for 15-min. Protein concentration of the supernatant was measured and equivalent sample amounts were subjected to SDS-PAGE. Protein was transferred to a PVDF membrane and blocked for 1-h in 5% skim milk. Membranes were incubated in the appropriate primary and secondary antibodies diluted in 1% skim milk or 3% BSA (see Supplementary Table 3 for catalog numbers and dilutions). Proteins were visualized using Amersham ECL Prime Western Blot Detection Reagent (VWR, CA89168-782) and a ChemiDoc Imaging System (Bio-Rad Laboratories).

### Immunofluorescence
Cells were seeded on no.1 glass coverslips (VWR, CA-89015725). For fixation, cells were washed with PBS and incubated in 3.7% paraformaldehyde (Electron Microscopy Sciences) for 15-min. When indicated, cells were treated with 0.025% digitonin for 10 s prior to fixation. Cells were washed twice with PBS and permeabilized with 0.1% Triton X-100 for 15-min. After two final PBS washes, cells were incubated in blocking buffer (10% FBS in PBS) for 1-h at room temperature. For mitochondrial staining, cells were fixed in warm 3.7% paraformaldehyde containing 5% sucrose. Cells were then incubated with the appropriate primary and secondary antibodies diluted in blocking buffer at room temperature for 1-h each (see Supplementary Table 3). For DAPI staining, cells were incubated in DAPI diluted 1:10,000 in PBS for 15-min. Coverslips were mounted on glass slides using DAKO mounting medium and stored at 4 °C. Fluorescence microscopy was performed on either a Zeiss LSM710 or LSM980 laser-scanning confocal microscope with a 63x/1.4 NA Plan-APOCRAMAT oil immersion objective. Images were taken using a combination of 6 laser lines (405,

458, 488, 514, 561, 633 nm) and the appropriate filters. *Z*-stacks were acquired using ZEN 2009 or ZEN 2019 (Zeiss Enhanced Navigation) software. For each experimental trial, all images were acquired using the same acquisition setting with minimal saturated pixel. Images used in figures were selected as representative images which best reflect the quantified data and were adjusted for brightness/contrast for presentation purposes only using Adobe Photoshop CS6.

### Quantification and statistical analyses

Immunofluorescence quantification analyses were conducted using Volocity software (Perkin Elmer, Version 6.3.1). Quantifications were generated by drawing a region of interest (ROI) around individual cells and applying the "Find Objects" algorithm in Volocity, using thresholding to control for size and intensity cut-off. Cell volume was recorded from the ROI. For peroxisome quantification, individual PMP70 puncta were counted and the number of PMP70 structures in a given ROI was then divided by cell volume. For ALIS quantification, FK1- or FK2-aggregates in a given ROI were identified based on applied thresholds, and the total volume of FK1- or FK2- aggregates was divided by cell volume to obtain relative aggregate volume. For mitochondria quantification, total fluorescence intensity of CVα was recorded and divided by cell volume. For αS quantification, total fluorescence intensity of αS was recorded and divided by cell volume. For colocalization analyses, Pearson's Correlation coefficients or Manders' coefficients were recorded from Volocity. For each experiment, at least three independent experiments (exact *n* provided in figure caption) were conducted, and mean values were calculated by averaging the appropriate measurements from 25 to 40 cells per independent experiment. All images are representative of *n* = 3–4 independent experiments, unless indicated otherwise in the legend.

Immunoblot densitometry quantifications were performed using ImageJ software and presented as an average of at least three (*n* = 3) independent experiments. Each protein band is normalized to its relevant probed loading control, either GAPDH, β-Actin, or Vinculin.

Statistical analyses were performed using either Student's *t*-test, or One- or Two-Way ANOVA followed by the appropriate post hoc test and are indicated in each figure caption. Statistical tests were performed using GraphPad Prism (Version 10.1.0) to include all available data for each experiment.

### Mitophagy assay

Parkin-dependent mitophagy was induced in GFP-Parkin HEK293 for 8-h with 2.5 μm oligomycin and 250 nM antimycin A1, and 2 μM rapamycin or 200 nM bafilomycin A1 where applicable. Cells were either lysed for immunoblotting, or fixed and immunostained for CVα, and analyzed with confocal fluorescent microscopy. Mitochondrial clearance was assessed by quantifying CVα relative fluorescence intensity, or calculating the percentage loss of mitochondrial proteins from densitometry quantifications of immunoblots.

Parkin-independent mitophagy was induced in human fibroblasts for 28-h with 1 mM 3-Hydroxy-1,2-dimethyl-4(1H)-pyridone (DFP), and 10 μM Carbonyl cyanide 3-chlorophenylhydrazone (CCCP) was used as a control. Cells were lysed for immunoblotting, or fixed and immunostained for HSP60 and LAMP1A. Mitophagy was detected in immunofluorescent images by an increase in HSP60 and LAMP1A colocalization (Manders' coefficient) and by scoring of mitophagy phenotype. Mitophagy phenotype was scored blind, and was defined as either mild, moderate, or severe based on the degree of mitochondrial collapse and colocalization with LAMP1A (see Supplementary Data Fig. 4 for examples of mitophagy phenotype scoring). Mitochondrial clearance was confirmed by calculating the percentage loss of mitochondrial proteins from densitometry quantifications of immunoblots.

### Aggrephagy assay

Cells were treated for 3-h with 5 μg mL$^{-1}$ puromycin to induce ALIS. Cells were washed with PBS and allowed a 5-h clearance period in DMEM, 2 μM rapamycin, or 20 μM LYN-1604 where applicable. Cells were fixed and immunostained for FK1 or FK2 and analyzed with confocal fluorescent microscopy. Aggrephagy was assessed by quantifying relative FK1/FK2-ALIS volume.

### αS clearance assay

αS PFF were generated using recombinant human αS expressing the A53T mutation associated with familial Parkinson's Disease. Briefly, 1 mg mL$^{-1}$ αS was incubated for 7 days at 37 °C with constant agitation in a buffer of 20 mM Tris-HCl, pH 7.4, and 100 mM NaCl, as previously described. Fibril formation was confirmed with a ThT fluorescence assay. For the αS clearance assay, monomeric (non-polymerized) αS or PFF were transfected for 4-h at a final concentration of 1 μg mL$^{-1}$ using Lipofectamine 2000, according to the manufacturer's protocol. Cells were washed with PBS and allowed an 8–15-h clearance period as indicated, in either DMEM, 2 μM rapamycin, or 0.25 mM Leupeptin and 2 μM E-64 where applicable. Cells were fixed and immunostained for αS, and analyzed with confocal fluorescent microscopy. αS clearance was assessed by quantifying αS relative fluorescent intensity in cells.

### Pexophagy assay

PMP34-GFP or PMP34-GFP-UBKo was expressed in cells for 48-h prior to fixation. For Torin1 experiments, cells were supplemented with 100 nM Torin1 for the final 24-h. Cells were either fixed and immunostained for PMP70 and analyzed with confocal microscopy or lysed for immunoblotting. Pexophagy was assessed by quantifying the percentage loss of PMP70 from PMP34-GFP to PMP34-GFP-UBKo conditions.

### Reporting summary

Further information on research design is available in the Nature Portfolio Reporting Summary linked to this article.

## Data availability

Source data for Figs. 1–9 and Supplementary Data Figs. 1–8 are provided as a Source Datafile. A reporting summary is also available in the Supplementary Information. All microscopy image datasets that support the findings of this manuscript are available from the corresponding authors upon request. Source data are provided with this paper.

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

## Acknowledgements

This work was supported by the Canada Institute of Health Research grants to P.K.K. (PJT #156196) and to R.B. (PJT# 156307), as well as the Ontario Graduate Scholarship, Hayden Hantho Award, and the Hilda and William Courtney Clayton Paediatric Research Fund to K.G.

## Author contributions

All authors contributed to the data analysis and interpretation of the data. K.G., R.B. and P.K.K. conceived the study. K.G., R.B. and P.K.K. designed the experiments. K.G. performed most of the experiments; R.W.L. and J.C.W. generated the αS fibrils and aided in the design of the αS fibrils experiments; L.F.D. performed the immunoblots in Fig. 8; K.G., R.B. and P.K.K. wrote the manuscript. All authors contributed to editing of the manuscript.

## Competing interests

The authors declare no competing interests.
