## [Peer Review File · Nature Communications]

Reviewers' Comments:

Reviewer #1:

Remarks to the Author:

The manuscript by Germain et al. investigates the possibility that upregulated pexophagy is sufficient to limit the capacity of selective autophagy in a range of cell culture models. The authors conclude that it does, through the exhaustion of the autophagy initiating factor ULK1. This work has been improved through the process of peer review, with the inclusion of more relevant experiments, and some improvements in the statistical analysis of the data, with one caveat described below.

However, while the study has been substantially improved, the overall impact of the work still remains limited by the limited nature of the question being addressed. Specifically, the authors ask whether induction of one form of selective autophagy is sufficient to impair another form of selective autophagy, and find that yes, this can happen because ULK1 becomes limiting. This finding is not a surprise, because upon induction of a single type of selective autophagy, not all cargos are turned over simultaneously – instead it can take hours for the cells to engulf and clear the targeted cargos. The obvious assumption is that one or more factors become limiting, and here the authors provide data indicating that ULK1 does in fact become limiting, although other factors may also become limiting as well. So while this work is a step forward in our understanding, it is a predictable one.

Much more interesting would have been to extend the work to explore the specifics of Zellweger Spectrum Disorder in more detail. Can the authors show in physiologically relevant models that the secondary inhibition of mitophagy or aggrephagy downstream of a peroxisome defect is causal for some or all of effects of the disorder? Or at least show that ULK1 depletion is relevant to the disease process in affected patients? If so, then the current study is more interesting and more impactful. Without these data, the work makes a solid but more limited contribution to our understand of the disorder.

Finally, the statistical analysis of the data has improved over the initial submission, but still requires further attention. Many of the statistical analyses of the data are performed with individual cells as data points, leading to inflated p-values. These analyses should be repeated using a more accepted definition of a biological replicate – see Lord et al. JCB 2020 for a detailed explanation.

Reviewer #2:

Remarks to the Author:

In this study, Dr. Germain and colleagues examined the effects of dysregulated pexophagy on aggrephagy and mitophagy, two other selective autophagy pathways, in HeLa, HEK293, and immortalized skin fibroblasts. They show that pexophagy-promoting conditions, such as PEX1 or PEX13 depletion, hinder the removal of α -synuclein aggregates and compromise membrane depolarization- and iron chelator-induced mitochondrial degradation. In addition, the authors provide evidence that neither the loss of PEX1 or PEX13, the impairment of peroxisome function, changes in mTORC1 activity caused by PEX1 or PEX13 depletion, the depletion of selective autophagy receptors, nor problems with autophagosome formation and flux can be attributed to the occurrence of these events. Instead, they demonstrate that reduced aggrephagy in cells with increased pexophagy is caused by “consumption” of the autophagy initiator ULK1, and that exogenous overexpression of this kinase or treatment with LYN-1604, a potent ULK1 activator, can reverse this impairment. Finally, by demonstrating reduced pexophagy in a Huntington’s disease cell model, the authors present another example of how increased substrate load can impose limitations on other types of selective autophagy. In summary, the work presented is of high quality and supports the idea that the degradative capacity of selective autophagy is restricted under baseline conditions. The physiological significance and potential clinical implications of these findings remain to be explored.

Specific comments

1. Lines 79-84 – The authors claim that PEX13 is not required for the initiation of peroxisome biogenesis. However, given two recent reports documenting that this peroxin exerts a key function in peroxisome biogenesis initiated by protein phase separation (doi: 10.1038/s41586-023-06044-1; doi: 10.1126/science.adf3971), this statement may be misleading.
2. Figure 1b – The authors claim that treatment of the cells with siPEX1 or siPEX13 induces pexophagy. However, this phenomenon is not evident in the PEX14 blot. Furthermore, considering that the observed differences in Figures 1c/1d are relatively minor, the quality of the manuscript would benefit profoundly by including immunoblots for PMP70.
3. Figure 1d – Were all corresponding "single depletion" and "ATG12 co-depletion data" statistically compared? That is, the differences between the "single depletion, siATG12" and "ATG12 co-depletion, siCTRL" conditions, which are expected to yield similar outcomes, seem to be larger than those of some other comparisons labeled with 4 stars. This observation holds true for the "single depletion, siPEX14" and "ATG12 co-depletion, siPEX14" data as well. Please clarify.
4. Figure 6B – Why does the anti-PEX1 staining not yield a visible band for PEX1-G843D, a mutation associated with a mild phenotype? This result is not in line with previously published data (see DOI: 10.1073/pnas.0914960107, Figure 4).
5. Extended Figure 4d – Please indicate a cell with a mild mitophagy phenotype.
6. Lines 439-460 – Throughout this study, the authors claim that an increase in pexophagy activity limits the capacity of the cell to clear damaged mitochondria and protein aggregates. However, if this were to be the main mechanism, how do the authors explain the accumulation of defective mitochondria in the absence of PEX5, the most frequently ubiquitinated peroxisomal protein (DOI: 10.1016/j.bbamcr.2014.11.017)? The authors propose a hypothesis suggesting that this phenomenon could be attributed to the presence of non-functional "ghost peroxisomes", which occur in the absence of PEX5. However, considering (a) the limited extent of pexophagy in PEX1- or PEX13-depleted cells, as observed in Figures 1b-d, and (b) the authors' assertive claim that this level of pexophagy already leads to substrate overload, the authors should experimentally document the effect of siPEX5 on mitophagy and aggregate clearance.
7. The authors are encouraged to provide a more comprehensive discussion regarding the physiological significance and potential clinical implications of their findings.

Reviewer #3:

Remarks to the Author:

The reviewer adequately addressed most of my concerns. However, there are still a few points that require their attention:

- 1) The second paragraph of their response to point #1 (on TBK1 and VPS34) should be included in the discussion. Also, I still think that analyzing WIPI2 and DFCP1 as only two markers for pre-autophagosomal structures in the course of their experimental paradigm will provide important mechanistic insights into the limitation of autophagosome biogenesis.
- 2) Regarding point #2: If the authors' hypothesis is true that "the number of autophagosomes the cell can generate at any given time is limited", then the upregulation of pexophagy (upon PEX1 or PEX13 knockdown) should be (at least partially) impaired during the clearance of protein aggregates (e.g. ALIS). The authors should test this reciprocal dependence of selective autophagy pathways to substantiate their conclusions.
- 3) Regarding point #3: How do the authors exclude that a-Syn does not reside inside lysosomes to which LC3 is lipidated (in a CARM dependent manner) due to membrane damage induced by a-Syn? The authors need to co-stain their a-Syn treated cells with a lysosomal marker.

4) The authors should carefully check the figure references throughout the text. In a number of places, the wrong figure is referenced (e.g. line 415.416). Also, there is no Extended Data Fig. 9.

Response to Reviewer's comments.

We thank the Reviewers for all of their insightful comments and suggestions, which have aided us in improving our manuscript. We describe below our responses to the three reviewers' comments and concerns. For convenience, we have used 'track changes' in our Word doc to help the reviewers identify changes to the revised manuscript.

Reviewer #1 (Remarks to the Author):

The manuscript by Germain et al. investigates the possibility that upregulated pexophagy is sufficient to limit the capacity of selective autophagy in a range of cell culture models. The authors conclude that it does, through the exhaustion of the autophagy initiating factor ULK1. This work has been improved through the process of peer review, with the inclusion of more relevant experiments, and some improvements in the statistical analysis of the data, with one caveat described below.

However, while the study has been substantially improved, the overall impact of the work still remains limited by the limited nature of the question being addressed. Specifically, the authors ask whether induction of one form of selective autophagy is sufficient to impair another form of selective autophagy, and find that yes, this can happen because ULK1 becomes limiting. This finding is not a surprise, because upon induction of a single type of selective autophagy, not all cargos are turned over simultaneously – instead it can take hours for the cells to engulf and clear the targeted cargos. The obvious assumption is that one or more factors become limiting, and here the authors provide data indicating that ULK1 does in fact become limiting, although other factors may also become limiting as well. So, while this work is a step forward in our understanding, it is a predictable one.

We agree with the reviewer that the idea of selective autophagy exhaustion in the context of an increased substrate load is intuitive. However, it has not been a hypothesis that has been commonly discussed or experimentally demonstrated in the autophagy field. Instead, in the context of autophagy in disease, the general thought is that selective autophagy is dysregulated. For example, in the field of neurodegenerative diseases, especially in proteinopathies where selective autophagy is most studied, the focus is largely on compromised autophagy and mitophagy due to aging or genetic mutations as a mechanism of disease development. To our knowledge, our study would be the first to demonstrate experimentally that one selective autophagy pathway can limit that of another pathway.

As the reviewer points out, in special situations, for example, during wholesale degradation of an entire network of mitochondria through simultaneous depolarization, there could be a limitation in one or more components of mitophagy. However, it is not obvious that ULK1 would be limited. The limited component(s) could be any at the level of mitophagy or autophagy. Further, there are other factors other than limited mitophagy/autophagy components that could impair the clearance of the entire mitochondrial network from a cell. Given the importance of mitochondria to eukaryotic cells, the slow turnover of global depolarized mitochondria can equally be interpreted to be caused by the negative regulator(s) of mitophagy in order to prevent the wholesale loss of mitochondria. To this point, there are several studies that provide evidence to suggest that mitophagy is regulated. During starvation,

mitochondria were found to become elongated in certain cells to prevent non-specific degradation (PMID: 21646527; 21478857). Further, cAMP-PKA negatively regulates both PINK1 and Parkin to prevent mitophagy (PMID: 27153535; 29692364). Finally, there are various deubiquitinating proteins and other post-translational modifying proteins that are associated with the regulation of mitophagy whose function during global depolarization have not been explored. Therefore, we argue that before our study, the assumption that one selective autophagy pathway could impede that of another pathway was not a foregone conclusion.

The aim of our study was to use a variety of models and substrates to empirically test and demonstrate that increased selective autophagy of one substrate can limit other selective pathways. Our study adds substantial value to the autophagy field by demonstrating and characterizing the limited nature of selective autophagy, which provides a foundation to ask how limited selective autophagy may contribute to several autophagy-related pathologies. For example, we have used two genes, PEX1 and PEX13, which are associated with the cerebo-hepato-renal disease, Zellweger Spectrum Disorder. The loss of these genes has been shown to cause peroxisome loss due to increased pexophagy. A common cellular phenotype of Zellweger Spectrum Disorder is the accumulation of damaged mitochondria and protein aggregates, but it is not understood why the various quality control systems cannot remove them efficiently. Our work now provides a possible mechanistic insight into their accumulation.

Much more interesting would have been to extend the work to explore the specifics of Zellweger Spectrum Disorder in more detail. Can the authors show in physiologically relevant models that the secondary inhibition of mitophagy or aggrephagy downstream of a peroxisome defect is causal for some or all of effects of the disorder? Or at least show that ULK1 depletion is relevant to the disease process in affected patients? If so, then the current study is more interesting and more impactful. Without these data, the work makes a solid but more limited contribution to our understand of the disorder.

We agree that exploring the specifics of Zellweger Spectrum Disorder is an interesting point, and it is an ongoing project in our group. We are currently investigating the role of limited selective autophagy and the potential of ULK1 as a therapeutic target in two animal models of Zellweger Spectrum Disorder, a Pex1-G844D mouse model (Heibler, S. *et al.*, Mol Genet Metab 2014, PMID: 24503136) and a PEX13-KO zebrafish model (Demers, N. *et al.*, Autophagy 2023, PMID: 36541703). These are multiyear studies that require generating multiple tissue specific double knockouts that are outside the scope of the present manuscript. However, these physiological studies would not have been possible without the current cell-based studies outlined in this manuscript as it provides the justification and hypothesis for our ongoing animal studies. Further, we believe that our findings will impact other fields to explore how limited selective autophagy may contribute to the pathophysiology of diseases such as Parkinson's and Huntington's. For this reason, we believe that our present manuscript would be of interest to the field of selective autophagy and its relationship to diseases. In Lines 698-705 of the discussion, we have expanded upon the relevance of our findings to Zellweger Spectrum Disorder.

Finally, the statistical analysis of the data has improved over the initial submission, but still requires further attention. Many of the statistical analyses of the data are performed with

individual cells as data points, leading to inflated p-values. These analyses should be repeated using a more accepted definition of a biological replicate – see Lord et al. JCB 2020 for a detailed explanation.

We have adjusted the graphs where statistical analyses were performed with individual cells as points (Fig. 1d,1i, Fig. 2b, Fig. 3g-i, Fig. 4i, and Supplemental Fig. 1b). These analyses were repeated using the means of each trial as points.

Reviewer #2 (Remarks to the Author):

In this study, Dr. Germain and colleagues examined the effects of dysregulated pexophagy on aggrephagy and mitophagy, two other selective autophagy pathways, in HeLa, HEK293, and immortalized skin fibroblasts. They show that pexophagy-promoting conditions, such as PEX1 or PEX13 depletion, hinder the removal of α -synuclein aggregates and compromise membrane depolarization- and iron chelator-induced mitochondrial degradation. In addition, the authors provide evidence that neither the loss of PEX1 or PEX13, the impairment of peroxisome function, changes in mTORC1 activity caused by PEX1 or PEX13 depletion, the depletion of selective autophagy receptors, nor problems with autophagosome formation and flux can be attributed to the occurrence of these events. Instead, they demonstrate that reduced aggrephagy in cells with increased pexophagy is caused by “consumption” of the autophagy initiator ULK1, and that exogenous overexpression of this kinase or treatment with LYN-1604, a potent ULK1 activator, can reverse this impairment. Finally, by demonstrating reduced pexophagy in a Huntington’s disease cell model, the authors present another example of how increased substrate load can impose limitations on other types of selective autophagy. In summary, the work presented is of high quality and supports the idea that the degradative capacity of selective autophagy is restricted under baseline conditions. The physiological significance and potential clinical implications of these findings remain to be explored.

Specific comments

1. Lines 79-84 – The authors claim that PEX13 is not required for the initiation of peroxisome biogenesis. However, given two recent reports documenting that this peroxin exerts a key function in peroxisome biogenesis initiated by protein phase separation (doi: 10.1038/s41586-023-06044-1; doi: 10.1126/science.adf3971), this statement may be misleading.

Lines 79-84 (now Lines 94-96) were changed to “since these two genes are not required for the formation of peroxisome membranes or membrane protein import, their loss results in the continuous degradation of old and new peroxisomes without causing significant cell death.” Additionally, lines 462-465 were changed to “cellular depletion of PEX13 or PEX1 (the most commonly mutated gene in ZSD) does not prevent the formation of **peroxisome membrane structures** but reduces their number by constantly degrading new peroxisomes resulting in upregulated pexophagy (Fig. 1).

2. Figure 1b – The authors claim that treatment of the cells with siPEX1 or siPEX13 induces pexophagy. However, this phenomenon is not evident in the PEX14 blot. Furthermore, considering that the observed differences in Figures 1c/1d are relatively minor, the quality of the manuscript would benefit profoundly by including immunoblots for PMP70.

We find that the depletion of PEX1 or PEX13 results in a decrease in peroxisome numbers, but this is not necessarily reflected by the total protein levels of peroxisomal membrane proteins. We believe that this is due to the difference in the half-life of peroxisomal membrane proteins versus the duration of the depletion assay. It was previously reported that peroxisomal membrane proteins have a half-life under 24 hours (Huybrechts, S.J. *et al*, *Traffic* 2009, PMID: 19719477). Since our siRNA experiments are typically 3-4 days post initial siRNA transfection, we find that most of the total membrane protein levels do not show a decrease as they target to residual peroxisomes. This results in peroxisomes that appear larger in immunofluorescence images of cells probed for peroxisome membrane proteins such as PMP70 (Fig. 3). This is likely due to a combination of expansion of the peroxisomal membrane (biogenesis) and to an increase in antibody labelling caused by increased level of membrane protein per peroxisomes. In addition, certain peroxins including PEX14 are known to mistarget to the mitochondria when peroxisomes are decreased, therefore total PEX14 protein levels may be unaffected in PEX1 or PEX13 depleted cells (Carmichael, R.E. *et al*, *Cells* 2022, PMID: 35741050; Nuebel, E. *et al*, *EMBO Rep* 2021, PMID: 34351705; Passmore, J.B. *et al*, *BBAMCR* 2020, PMID: 32224193). In contrast loss of PEX1 or PEX13 results in decreased total levels of peroxisomal matrix proteins (Law, K.B. *et al*, *Autophagy* 2017, PMID: 28521612; Demers, N.D. *et al*, *Autophagy* 2023, PMID: 36541703). The data in Figures 1b-d serve as a point of reference and are not meant to provide extensive examination of PEX1 and PEX13 in pexophagy as we have already characterized this in detail in our previous reports (Law, K.B. *et al*, *Autophagy* 2017, PMID: 28521612; Demers, N.D. *et al*, *Autophagy* 2023, PMID: 36541703). Instead, these data are intended to serve as a reference point to aid in the readability of the manuscript.

3. Figure 1d – Were all corresponding "single depletion" and "ATG12 co-depletion data" statistically compared? That is, the differences between the "single depletion, siATG12" and "ATG12 co-depletion, siCTRL" conditions, which are expected to yield similar outcomes, seem to be larger than those of some other comparisons labeled with 4 stars. This observation holds true for the "single depletion, siPEX14" and "ATG12 co-depletion, siPEX14" data as well. Please clarify.

These statistical comparisons were excluded from the graph to minimize clutter as they were not essential information for the questions being asked. However, to avoid confusion in our revised manuscript we have added in all of the statistical comparisons for Fig. 1d. Please note these analyses were revised according to Reviewer 1's request to change from using individual cells as data points to using means as data points.

4. Figure 6B – Why does the anti-PEX1 staining not yield a visible band for PEX1-G843D, a mutation associated with a mild phenotype? This result is not in line with previously published data (see DOI: 10.1073/pnas.0914960107, Figure 4).

The lack of visible PEX1 band from the PEX1-G843D lysate is due to the lower level of PEX1-G843D expression as PEX1-G843D is less stable than the wild-type protein (Walter, C. *et al*, *Am J Hum Genet* 2001, PMID: 11389485). This can be seen in published data where PEX1 protein levels are visibly decreased in PEX1-G843D cells compared to wild type (Zhang, R. *et al*, *PNAS* 2010, PMID: 20212125; Fig. 4a lane C vs. lane 1). In our immunoblot, we elected to show a less exposed image that is more amenable to quantification.

5. Extended Figure 4d – Please indicate a cell with a mild mitophagy phenotype.

We have adjusted Supplemental Figure 3d to indicate a mild mitophagy phenotype.

6. Lines 439-460 – Throughout this study, the authors claim that an increase in pexophagy activity limits the capacity of the cell to clear damaged mitochondria and protein aggregates. However, if this were to be the main mechanism, how do the authors explain the accumulation of defective mitochondria in the absence of PEX5, the most frequently ubiquitinated peroxisomal protein (DOI: 10.1016/j.bbamcr.2014.11.017)? The authors propose a hypothesis suggesting that this phenomenon could be attributed to the presence of non-functional "ghost peroxisomes", which occur in the absence of PEX5. However, considering (a) the limited extent of pexophagy in PEX1- or PEX13-depleted cells, as observed in Figures 1b-d, and (b) the authors' assertive claim that this level of pexophagy already leads to substrate overload, the authors should experimentally document the effect of siPEX5 on mitophagy and aggrephagy.

As functional peroxisomes are required for mitochondrial health (Fransen, M., Lismont, C., Walton, P., *Int J Mol Sci* 2017, PMID: 28538669), and PEX5 deficiency results in severe peroxisomal defects, the accumulation of defective mitochondria in the absence of PEX5 is unsurprising. In physiological conditions, this could result from limited autophagy due to excess "ghost peroxisomes", or changes in global autophagy as the loss of PEX5 has been shown to disrupt autophagy through multiple axes including mTORC1 regulation (Bhandari, S. et al, *Cell Mol Life Sci* 2023, PMID: 36821008; Eun, S.Y. et al, *Exp Mol Med* 2018, PMID: 29622767; Zhang, J. et al, *Nat Cell Biol* 2015, PMID: 26344566). However, previous reporting has demonstrated that a two-to-three-day knockdown of PEX5 does not upregulate pexophagy in cultured cells under basal conditions, therefore we would not expect it to acutely limit selective autophagy (Demers, N.D. *et al*, *Autophagy* 2023, PMID: 36541703). We performed one trial of our puromycin aggrephagy assay with siPEX5 to test whether it would affect aggrephagy (see data below) and found that there was no change to aggrephagy capacity compared to control conditions. Future studies from our lab will perform a comprehensive examination of the effects of PEX5 on autophagy as it pertains to Zellweger Spectrum Disorder and whether "ghost peroxisomes" are degraded through autophagy in these conditions. However, this is outside the scope of our present study that is focused on the effects of acute upregulation of pexophagy.

(a) HeLa cells were treated with the indicated siRNA and subjected to the puromycin aggregophagy assay. Representative images display cells at each stage in the assay: DMEM, 3-h $5\mu\text{g mL}^{-1}$ Puromycin, or 3-h $5\mu\text{g mL}^{-1}$ Puromycin followed by 5-h clearance period in DMEM. Cells were immunostained with the ubiquitin antibody FK2. **(b)** Quantification of aggregate volume, as seen in manuscript. Data indicates the mean value from 30 cells per condition.

7. The authors are encouraged to provide a more comprehensive discussion regarding the physiological significance and potential clinical implications of their findings.

We have expanded upon the relevance of our findings to Zellweger Spectrum Disorder in the end of our Discussion:

“Using a ZSD model, we provide here evidence to support that distinct selective autophagy pathways can influence each other, and that the degradative capacity of selective autophagy can be acutely limited by an increased substrate load in cells. Our studies provide justification for targeting the activation of selective autophagy or inhibition of pexophagy as a strategy for ZSD treatment. However, such studies must weigh how such activation/inhibition can affect different tissues and cell types. Autophagy is highly regulated, and it is not known how long-term activation of autophagy affects neurons or glia. Similarly, it is not clear what effect an accumulation of damaged or partially assembled peroxisomes has on the regulation of both lipid and redox homeostasis, which could impact autophagy, therefore future studies of

pexophagy in ZSD model systems are required. Further, it remains to be determined whether a similar mechanism of limited autophagy capacity caused by an increase in one specific substrate contributes to the cellular pathophysiology of other diseases, including proteinopathy and mitochondrial disease.

Reviewer #3 (Remarks to the Author):

The reviewer adequately addressed most of my concerns. However, there are still a few points that require their attention:

1) The second paragraph of their response to point #1 (on TBK1 and VPS34) should be included in the discussion. Also, I still think that analyzing WIPI2 and DFCP1 as only two markers for pre-autophagosomal structures in the course of their experimental paradigm will provide important mechanistic insights into the limitation of autophagosome biogenesis.

We amended our manuscript to include TBK1 and VPS34 in our discussion (Lines 640-643).

We agree with the reviewer that quantifying pre-autophagosome structures would have added to this study. However, we have found such studies difficult due to the limitations of our detection assays, the rate of autophagosome biogenesis in basal conditions (non-amino acid starved conditions), and the short lifespan of pre-autophagosomes. There are two main caveats to monitoring preautophagosomal structures in our assays. First, overexpression of early autophagy factors may influence autophagosome formation dynamics in our assays, as was observed with ULK1 overexpression in Fig. 9 and Supplemental Data Fig. 8. Second, expression levels may differ between cells with transient transfection leading to noisy data. To overcome these caveats, we aimed to examine endogenous levels of preautophagosomal markers by immunofluorescent imaging. However, despite our efforts, we could not find a commercially available WIPI2 or DFCP1 antibody that worked well enough for robust quantification of preautophagosomal structures to mark pre-autophagosomes in basal conditions. Such structures were only visible in cells where global autophagy was induced by mTORC1 inhibition. We also attempted to monitor preautophagosomal markers using a commercially available ATG16L antibody (see figure below) but again were not able to observe the formation of discernable puncta in our selective autophagy assays without global autophagy induction (see data below). It is, therefore, our aim for our next study to generate Halo-Tagged CRISPR knock-in cell lines tagging various early autophagy proteins, including ULK1, VPS34, TBK1, ATG2 and ATG9, to monitor their recruitment dynamics and preautophagosomal structure formation during increased selective autophagy demand.

Figure: Immunofluorescent images of HeLa after 3 h with Puromycin. The cells were probed with antibodies the ubiquitin antibody FK2 (white) and against ATG16L (green). We were not able to observe phagophores with the ATG16L antibody.

2) Regarding point #2: If the authors' hypothesis is true that “the number of autophagosomes the cell can generate at any given time is limited”, then the upregulation of pexophagy (upon PEX1 or PEX13 knockdown) should be (at least partially) impaired during the clearance of protein aggregates (e.g., ALIS). The authors should test this reciprocal dependence of selective autophagy pathways to substantiate their conclusions.

We thank the reviewer for this excellent point. Unfortunately, testing the effects of ALIS clearance on pexophagy is difficult due to the 8-h time frame between ALIS formation and clearance, which we found too short to detect a robust change in pexophagy that typically requires at least 24-48 hours to observe significant changes in peroxisome levels with PEX1 or PEX13 depletion. Therefore, we carried out a new experiment to test the reciprocal dependence of selective autophagy pathways to substantiate our conclusions. In this experiment (new figure 7), we turned to alpha-Synuclein (aS) aggregates that take between 12-20-h to form and clear to ask whether it can reduce pexophagy.

Specifically, we asked whether aggrephagy affected peroxisome loss induced by PEX13 depletion. We selected PEX13-induced pexophagy for these experiments as PEX13 is depleted 20-h after a single siRNA transfection (Fig. 7a) and results in significant peroxisome clearance 39-h after siRNA transfection (Fig. 7e: siPEX13 Mon, 4h aS +15h Clear). We induced aggrephagy by aS PFF transfection into cells 20-h after a single PEX13 siRNA transfection and fixed cells 19-h after aS PFF transfection (39-h after PEX13 siRNA) (Fig. 7b). As a control, we transfected monomeric aS that does not produce aggregates in cells. PEX13-depleted cells were able to reduce aS aggregates by 48% after a clearance period, which was significantly less than control siRNA-depleted cells. To test whether aggrephagy influenced pexophagy, we quantified PMP70-immunostained puncta throughout our assay (Fig. 7c,e). In PEX13-depleted cells transfected with monomeric aS, peroxisomes were significantly reduced by the end of the 39-h assay,

supporting upregulated pexophagy (Fig. 7c,e). However, a significant loss of peroxisomes was not observed in PEX13-depleted cells transfected with aS PFF (Fig. 7c,e), suggesting that increased aggrephagy acutely impairs pexophagy.

These data suggest that akin to the ability of upregulated pexophagy to impair aggrephagy, upregulated aggrephagy can reciprocally impair pexophagy. Further, our experiments in a cell model of Huntington's Disease (Fig. 8, Supplemental Data Fig. 5, Supplemental Data Fig. 6) also addressed the reciprocal dependence of selective autophagy pathways. In these assays we observed that excess aggrephagy due to mutant HTT expression decreased pexophagy in striatal cells, supporting that pexophagy can similarly be limited by increased substrate load. Together, these studies strongly suggest that aggrephagy can reduce pexophagy.

3) Regarding point #3: How do the authors exclude that a-Syn does not reside inside lysosomes to which LC3 is lipidated (in a CASM dependent manner) due to membrane damage induced by a-Syn? The authors need to co-stain their a-Syn treated cells with a lysosomal marker.

To address this point, we repeated our a-Syn assay in cells expressing a lysosomal marker LAMP1A-mEmerald (Fig. 5b) with a control condition where lysosomal degradation is prevented using the lysosomal protease inhibitors Leupeptin + E-64. We also note that all of our a-Syn assays transfect a-Syn PFF into cells using Lipofectamine which should circumvent a-Syn internalization by phagocytosis and CASM. However, this does not exclude that CASM may mediate a-Syn clearance in other experimental setups where a-Syn is added directly to the culture medium without a transfection agent and internalized via phagocytosis.

4) The authors should carefully check the figure references throughout the text. In a number of places, the wrong figure is referenced (e.g. line 415.416). Also, there is no Extended Data Fig. 9.

We thank the reviewer for noticing these errors, we have carefully checked that the figures are all appropriately referenced in the revised manuscript.

Reviewers' Comments:

Reviewer #2:

Remarks to the Author:

The authors have adequately addressed my concerns and questions.

Reviewer #3:

Remarks to the Author:

The authors have sufficiently answered all questions and concerns. No further revisions are requested.

REVIEWERS' COMMENTS

Reviewer #2 (Remarks to the Author):

The authors have adequately addressed my concerns and questions.

Reviewer #3 (Remarks to the Author):

The authors have sufficiently answered all questions and concerns. No further revisions are requested.

We thank all the reviewers for their suggestions that helped improve our manuscript and insights that will direct our future investigations.